# 🫘 PINTO: Faithful Language Reasoning Using Prompt-Generated Rationales

**Peifeng Wang**[1,2]**, Aaron Chan**[1]**, Filip Ilievski**[1,2]**, Muhao Chen**[1,2]**, Xiang Ren**[1,2]
[1]Department of Computer Science, University of Southern California
[2]Information Sciences Institute, University of Southern California
`{peifengw, chanaaro, muhaoche, xiangren}@usc.edu, ilievski@isi.edu`

## Abstract

Neural language models (LMs) have achieved impressive results on various language-based reasoning tasks by utilizing latent knowledge encoded in their own pretrained parameters. To make this reasoning process more explicit, recent works retrieve a rationalizing LM's internal knowledge by training or prompting it to generate free-text rationales, which can be used to guide task predictions made by either the same LM or a separate reasoning LM. However, rationalizing LMs require expensive rationale annotation and/or computation, without any assurance that their generated rationales improve LM task performance or faithfully reflect LM decision-making. In this paper, we propose PINTO, an LM pipeline that *rationalizes* via prompt-based learning, and learns to *faithfully reason over rationales* via counterfactual regularization. First, PINTO maps out a suitable reasoning process for the task input by prompting a frozen rationalizing LM to generate a free-text rationale. Second, PINTO's reasoning LM is fine-tuned to solve the task using the generated rationale as context, while regularized to output less confident predictions when the rationale is perturbed. Across four datasets, we show that PINTO significantly improves the generalization ability of the reasoning LM, yielding higher performance on both in-distribution and out-of-distribution test sets. Also, we find that PINTO's rationales are more faithful to its task predictions than those generated by competitive baselines.[1]

## 1 Introduction

Many language-based reasoning tasks require retrieving and reasoning over knowledge beyond the task input—*e.g.,* commonsense reasoning and closed-book QA (Fig. 1, left) (Talmor et al., 2018; Mihaylov et al., 2018). Neural language models (LMs) have achieved impressive results on such tasks by utilizing latent knowledge encoded in their pretrained parameters (Raffel et al., 2020b; Brown et al., 2020). Still, given LMs' black-box nature, it is unclear whether this knowledge is being used properly (Doshi-Velez & Kim, 2017; Lipton, 2018). Previous studies have shown that LMs often learn spurious correlations from artifacts in downstream training data, thus limiting their generalizability (Branco et al., 2021; Geirhos et al., 2020; D'Amour et al., 2020).

With this in mind, a number of prior works aim to make LMs' reasoning processes more *explicit* by generating free-text rationales, which use LMs' internal knowledge to describe a reasoning process in natural language (Narang et al., 2020; Wei et al., 2022b; Marasović et al., 2022; Zelikman et al., 2022). In the *fine-tuned self-rationalizing* paradigm, a single LM is fine-tuned to jointly generate the task output and rationale (Narang et al., 2020; Marasović et al., 2022; Zelikman et al., 2022). In the *prompted self-rationalizing* paradigm, a single LM is instead frozen and prompted to jointly generate the task output and rationale, with the prompt consisting of a few input-output-rationale demonstrations (Wei et al., 2022b). In the *pipeline-rationalizing* paradigm, a fine-tuned rationalizing LM first generates the rationale, which is then used as input for a separate fine-tuned reasoning LM to generate the output (Kumar & Talukdar, 2020; Rajani et al., 2019).

---

[1]Code and data used in our experiments can be found at `https://github.com/wangpf3/pinto-faithful-language-reasoning`.

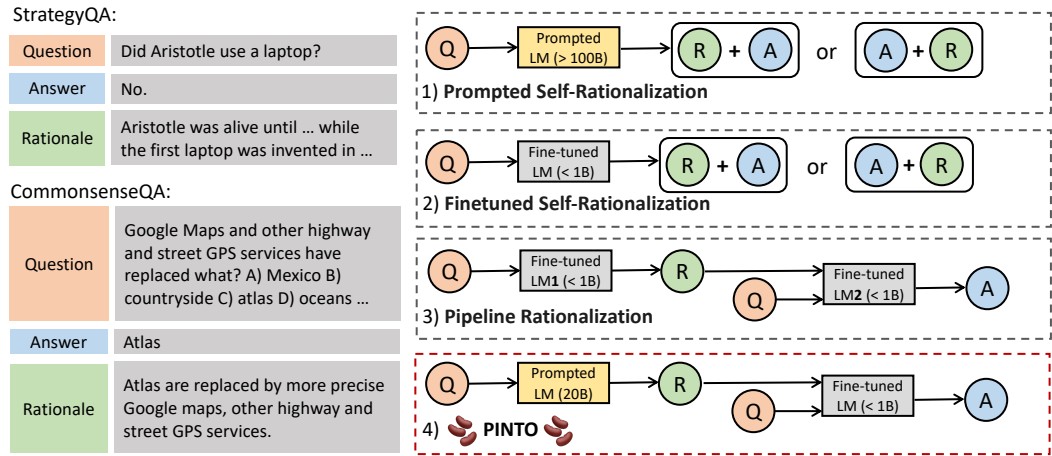

Figure 1: **Rationale-Based Language Reasoning.** (a) Examples of reasoning tasks that require implicit knowledge beyond task inputs. (b) Comparison of existing paradigms for providing free-text rationales along with predictions.

However, when considering generalization performance, reliability, and deployment costs, these existing paradigms all have key limitations. Fine-tuned self-rationalizing LMs often perform worse than non-rationalizing LMs, since their parameters are learned using two relatively dissimilar objectives, while also requiring expensive rationale annotations (Wiegreffe et al., 2020; Narang et al., 2020). Prompted self-rationalizing LMs yield strong task performance and only need a few rationale demonstrations for the prompt, but are computationally prohibitive since they generally require very large-scale (*i.e.,* over 100B parameters) LMs to work effectively (Wei et al., 2022a;b). Besides requiring expensive rationale annotations, pipeline-rationalizing LMs' generated rationale forms a non-differentiable bottleneck between the two modules, which complicates end-to-end training and can hurt task performance (Wiegreffe et al., 2020; Hase et al., 2020). Moreover, none of these paradigms has a mechanism for regularizing the rationale generation to *faithfully* reflect the reasoning process of the LM, without hurting task performance.

In this paper, we propose **P**rompted Rat**I**onalizing with Cou**NT**erfactual Reas**O**ning (🫘 **PINTO**), an LM pipeline that rationalizes via prompt-based learning, then reasons over the task input and rationale via counterfactual regularization. PINTO's *rationalizing module* is a medium-scale (*i.e.,* 20B parameters) LM that contains vast latent knowledge obtained via pretraining (Black et al., 2022). Though prohibitive to fine-tune, it is affordable for prompt-based learning. Given the task input and a minimal input-output-rationale demonstration prompt, the rationalizing module uses its internal knowledge to map out a suitable reasoning process for the task input by generating a free-text rationale. The rationalizing module is frozen during fine-tuning, which drastically reduces training costs and prevents it from exploiting spurious shortcuts in the downstream training data. PINTO's *reasoning module* is a small-scale (*i.e.,* under 1B parameters) LM to which knowledge is transferred from the rationalizing module. The reasoning module is fine-tuned to solve the downstream reasoning task by using the generated rationale as context for the task input. Crucially, to help ensure that the reasoning module's behavior is dictated by the rationale (instead of by spurious shortcuts), the reasoning module is regularized to output less confident predictions when the rationale is noisily perturbed. To simulate shortcut reasoning, we consider two rationale perturbation strategies: token masking (*i.e.,* rationale is ignored) and token replacement (*i.e.,* rationale is misused).

Across four question answering datasets (CSQA, StrategyQA, OpenBookQA, QASC), we show that PINTO significantly improves the reasoning LM's generalization, yielding higher performance on both in-distribution (ID) and out-of-distribution (OOD) test sets. Also, we find that rationales are utilized more faithfully by PINTO than by other methods, leading to better performance in low-resource settings. Furthermore, we show that PINTO's counterfactual regularization allows us to further improve task performance with refined rationales.

## 2   RATIONALE-BASED LANGUAGE REASONING

In this work, we study LMs' ability to reason about language using implicit knowledge. We consider a specific type of multi-choice question answering (QA) tasks where the required knowledge for

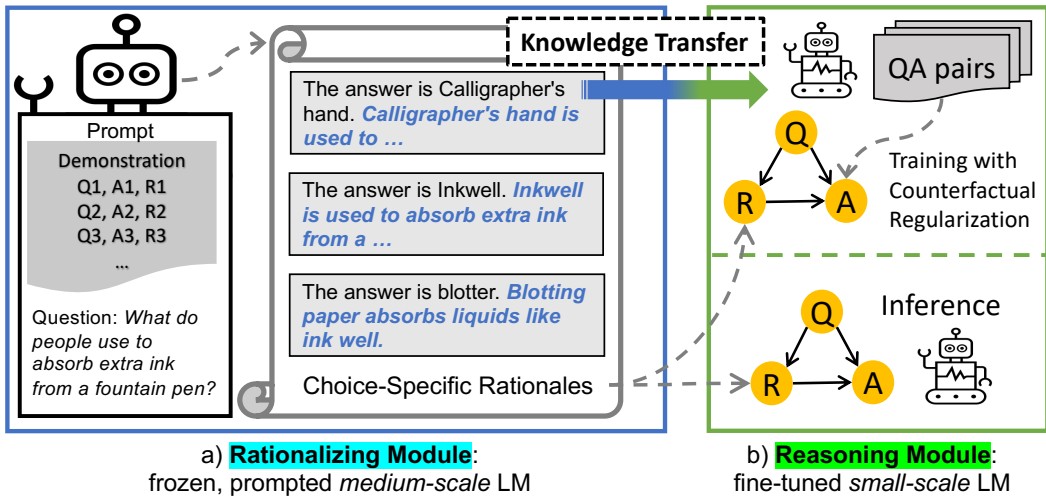

**Figure 2: Overview of PINTO.** (1) A frozen medium-scale LM is prompted to generate choice-specific rationales. (2) A small-scale LM is fine-tuned to reason over the generated rationales. (3) We introduce counterfactual regularization in addition to standard training loss to ensure the rationales are leveraged properly. During inference, the rationalizing LM is prompted with a new question to generate rationales, which are provided to the reasoning module to make a prediction.

answering the question is not explicitly provided in the input and needs to be inferred from the LM's parameters (Talmor et al., 2019; Khot et al., 2020): Given a question $q$ and a set of answer choices $A = \{a_i\}$, the model's goal is to predict a plausibility score $\rho(q, a_i)$ for each $(q, a_i)$ pair, so that the predicted answer $\hat{a} = \arg\max_{a_i \in A} \rho(q, a_i)$ matches the correct answer choice $a^* \in A$.

Motivated by LMs' common tendency to exploit reasoning shortcuts when solving tasks (Branco et al., 2021), we focus on methods that explicitly generate free-text rationales to explain their predictions. Whereas extractive rationales are limited to input token scoring (Denil et al., 2014; Sundararajan et al., 2017; Chan et al., 2022), free-text rationales use natural language to describe a reasoning process (*e.g.,* with knowledge beyond the task input) (Narang et al., 2020; Wei et al., 2022b). Below, we discuss several paradigms (see also Fig. 1) for rationale-based language reasoning.

**Fine-Tuned Self-Rationalization** In this paradigm, an LM is *fine-tuned* to autogregressively generate the task output and rationale as a single sequence (Narang et al., 2020; Liu et al., 2018). If the rationale is generated after the task output, then the rationale is conditioned on the task output, and vice versa. Since the LM parameters are shared across two relatively dissimilar objectives, they often perform worse than non-rationalizing LMs (Wiegreffe et al., 2020; Narang et al., 2020). Notably, this paradigm requires expensive rationale annotations for all training instances.

**Prompted Self-Rationalization** In this paradigm, a pretrained LM is *frozen* and *prompted* to autogregressively generate the task output and rationale as a single sequence, with the prompt consisting of a few input-output-rationale demonstrations (Lampinen et al., 2022; Wei et al., 2022b). If the rationale is generated after the task output, then the rationale is conditioned on the task output, and vice versa. This paradigm performs well and only needs a few rationale annotations for the prompt, but it is computationally prohibitive since it generally requires very large-scale (*i.e.,* over 100B parameters) LMs to work effectively (Lampinen et al., 2022; Wei et al., 2022b).

**Pipeline Rationalization** In this paradigm, a fine-tuned rationalizing LM first generates the rationale, which is then used as input for a separate fine-tuned reasoning LM to predict the task output (Kumar & Talukdar, 2020; Rajani et al., 2019). Here, the generated rationale forms a discrete (*i.e.,* non-differentiable) bottleneck between the two modules, which complicates end-to-end training and can hurt task performance (Wiegreffe et al., 2020; Hase et al., 2020). Additionally, the dedicated rationalizing LM requires extra rationale annotation/computation costs.

## 3 PINTO: Faithful Language Reasoning

PINTO is a two-stage, rationalize-then-reason pipeline, designed to address the limitations of existing paradigms for rationale-based language reasoning (§2). Like the pipeline rationalization paradigm, PINTO has separate modules for rationalizing and reasoning (Fig. 2). However,

Table 1: **Rationalization Prompts.** The format of our prompts for rationalization with a medium-scale LM. The prompt consists of a few examples as demonstration on how to rationalize for a question-choice pair and placeholders for new question and a target choice.

| Task | CommonsenseQA | OpenBookQA |
|------|---------------|------------|
| Prompt | **Q**: What do people use to absorb extra ink from a fountain pen? 
 **Answer Choices**: (a) shirt pocket (b) calligrapher's hand (c) inkwell (d) desk drawer (e) blotter 
 **A**: The answer is blotter. *Blotting paper absorbs liquids like ink well.* | **Q**: How do you reduce pollution? 
 **Answer choices**:(a) igniting fuel and oxidiser (b) transportation technology ... (h) using less resources 
 **A**: The answer is using less resources. *Conserving resources has a positive impact on the environment. Use of resources affects the environment such as pollution.* |

PINTO's rationalizing module is prompted instead of fine-tuned. Thus, PINTO does not suffer from the non-differentiable bottleneck issue and has lower rationale annotation/computation costs.

Following prior works, PINTO is based on choice-specific rationales (Kumar & Talukdar, 2020; Hase et al., 2020). First, given $q$ and $A$, the *rationalizing module* generates a set of choice-specific rationales $R = \{r_i\}$, where each $r_i$ explains a reasoning process that supports answer choice $a_i \in A$ (§3.1), as opposed to generating one rationale per question. We opt for this design choice because rationales are often answer-leaking (Sun et al., 2022), *i.e.,* the rationale itself is already sufficiently predictive of one of the answer choices. If the rationalizing module only generates one rationale per question, then it is forced to make an "early decision" on the predicted answer, such that the reasoning module would only be left to recover the answer from the rationale (Kumar & Talukdar, 2020). While prior works require expensive rationale annotations to train/prompt the rationalizing module (Kumar & Talukdar, 2020; Hase et al., 2020), PINTO's rationalizing module is a frozen pretrained LM that uses only a few question-answer-rationale demonstrations as a prompt (§3.1). Second, given $q$, $a_i \in A$, and $r_i \in R$, the *reasoning module* outputs plausibility score $\rho(q, a_i, r_i)$ (§3.2). We also design a regularization objective that encourages the reasoning module to properly use the rationales to predict the answer (§3.3). We describe each module in more detail below.

### 3.1 RATIONALIZING MODULE

Prior works mainly rely on human-annotated rationales for teaching a model to rationalize (Kumar & Talukdar, 2020; Hase et al., 2020; Sun et al., 2022). However, such rationale annotations are expensive and frequently of low quality (Aggarwal et al., 2021; Sun et al., 2022; Rajani et al., 2019), *e.g.,* not providing sufficient knowledge to support a given answer. Meanwhile, a recent study shows that rationales automatically generated by pretrained LMs are often preferable over human-annotated rationales (Wiegreffe et al., 2021). Therefore, for PINTO's rationalizing module, we propose using a pretrained LM to generate rationales via in-context learning, which prompts the frozen LM to retrieve knowledge from its parameters (Wei et al., 2022b).

The prompt consists of a fixed set of question-answer-rationale demonstrations that are randomly selected from the training set. Each demonstration consists of a question $q$, answer choices $A$,[2] gold answer $a^* \in A$, and a human-annotated free-text rationale $r^* \in R$ for $a^*$ (Table 1).[3] With this prompt $p$, we use the LM to generate rationales for every instance from the dataset. Specifically, for each $a_i \in A$ of some instance $(q, A)$, the rationalizing LM's input is constructed as $[p, q, A, a_i]$. Then, we use greedy decoding of the LM output to obtain rationale $r_i$ for $a_i$. Note that the LM input does not have any information about the gold answer $a^*$. Our rationalizing module's design assumes that $r_i$ will be aligned with accurate knowledge if and only if $a_i = a^*$, since it should intuitively be difficult to retrieve correct knowledge that supports an incorrect answer choice (see Table 11 in the appendix for examples of the generation). The reasoning module then predicts the correct answer by reasoning over the rationales for each answer choice.

### 3.2 REASONING MODULE

Given a question $q$, the answer choices $A$, answer candidate $a_i \in A$, and rationale $r_i$, the reasoning module learns to output plausibility score $\rho_i = \rho(q, A, a_i, r_i)$. Following prior works, we use

---

[2]We include the answer choices $A$ in the prompt so that the LM is aware of all the available choices and thus could generate a rationale that is more distinctive.

[3]As opposed to full human annotation, we only need a few (usually $< 8$) examples per dataset.

a text-to-text Transformer LM as the backbone of our reasoning module (Wiegreffe et al., 2020; Hase et al., 2020). For each $a_i$, the reasoning module's input is defined as the token sequence $s = [q \oplus a_1 \oplus ... \oplus a_{|A|} \oplus r_i]$, where $\oplus$ denotes concatenation. Meanwhile, the reasoning module's output is obtained by sequentially teacher-forcing $a_i$'s tokens $t_i = [t_i^1, t_i^2, ..., t_i^{|a_i|}]$ into the decoder, rather than via greedy decoding. This way, we can compute the reasoning module's output token probabilities for arbitrary answer choices $a_i$. Following Shwartz et al. (2020), we compute $a_i$'s plausibility score $\rho_i$ by aggregating the probabilities $P$ of tokens $t_i^j$ as:

$$\rho_i = \frac{1}{|a_i|} \sum_{j=1}^{|a_i|} \log P(t_i^j \mid t_i^{j-1}, ..., t_i^2, t_i^1, q, A, r_i).$$

Next, we use the softmax function to normalize $\rho_i$ as probability $P(a_i \mid q, A, R) = e^{\rho_i} / \sum_{j=1}^{|A|} e^{\rho_j}$. During inference, given question $q$ and answer choices $A$, the rationalizing module first generates rationales $R = \{r_i\}$, then the reasoning module computes the predicted answer choice as $\hat{a} = \arg\max_{a_i \in A} P(a_i \mid q, A, R)$.

### 3.3 TRAINING

For multi-choice QA, the standard training objective is to maximize the likelihood of the correct answer choice using cross-entropy loss, computed as:

$$\mathcal{L}_{\text{std}} = - \sum_{a_i \in A} Q(a_i \mid q, A) \log P(a_i \mid q, A, R), \tag{1}$$

where $Q(a_i \mid q, A)$ is 1 if $a_i = a^*$ and 0 otherwise. Let $Q(A \mid q, A)$ be the one-hot target distribution over all $a_i \in A$. There can be spurious correlations between $q$ and $A$ (Branco et al., 2021), so the reasoning module may take undesirable shortcuts instead of properly using the rationale to predict the answer (Gururangan et al., 2018; McCoy et al., 2019). In this case, the rationales would be unfaithful in explaining the model's behavior and useless for model debugging.

To address this, we introduce a counterfactual regularization objective in which the reasoning module is regularized to output less confident predictions when the rationale is not utilized properly (*i.e.,* shortcuts are used). This is implemented using label smoothing (Szegedy et al., 2016), which softens the target distribution $Q(A \mid q, A)$ by linearly combining it with a noisy distribution $U(A \mid q, A)$, often set as the uniform distribution. Therefore, given tunable label smoothing factor $0 < \epsilon < 1$, we compute the label-smoothed target distribution as: $Q'(A \mid q, A) = (1 - \epsilon) Q(A \mid q, A) + \epsilon U(A \mid q, A)$.

In order to simulate shortcut reasoning, we consider two strategies for perturbing the generated rationales $r_i$. **Token Masking** addresses the case where the reasoning module ignores the rationale and instead exploits spurious cues in the rest of the input. To simulate this, we mask out the rationales in the input. Recall that the backbone of the reasoning module is a Transformer LM, which uses a self-attention mechanism to aggregate information across tokens. Hence, we implement rationale masking by zeroing the attention mask for rationale tokens.[4] **Token Replacement** addresses the case where the reasoning module misunderstands the rationales' meaning and thus uses them improperly. To simulate this, we randomly replace $k\%$ of the rationale tokens with other tokens uniformly sampled from the entire language vocabulary.

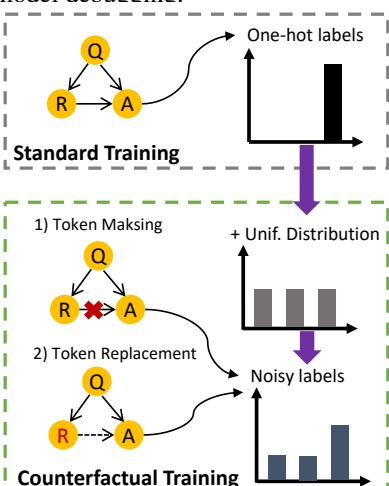

Figure 3: **Standard Training vs. Counterfactual Training.** For counterfactual regularization, we train the reasoning module with noisy labels when the rationale tokens are either masked or replaced.

At each fine-tuning step, we randomly select one of the strategies for obtaining perturbed rationales $R' = \{r_i'\}$, which helps keep the LM from overfitting to any particular strategy. Then, the

---

[4]We do not choose to replace the tokens in a rationale with special mask tokens since the LM is already pretrained to recover the mask tokens, and we want to ensure that this ability is completely deprived.

counterfactual regularization loss is computed as:

$$\mathcal{L}_{\text{c-reg}} = -\sum_{a_i \in A} Q'(a_i \,|\, q, A) \log P(a_i \,|\, q, A, R').$$ (2)

This counterfactual regularization teaches the reasoning module to be less confident when the rationales are either absent or problematic, so that it can learn to make sounder use of the rationales.

## 4 EXPERIMENTAL SETUP

**Questions and hypotheses** We design experiments to answer the following questions: (1) What is the impact of our PINTO pipeline on faithfulness and end-task performance? We expect our pipeline with counterfactual training technique to obtain improvements in both aspects. (2) How does the quality of rationales affect the end-task performance of PINTO? We hypothesize that improving the quality of the rationales of PINTO improves its accuracy. (3) Does faithful reasoning based on rationales lead to better generalization? We expect that a method like PINTO that learns to rely on rationales can better generalize to a low resource setting and out-of-distribution (OOD) datasets.

**Datasets** We experiment with several CSR benchmarks. (1) CommonsenseQA (Talmor et al., 2018) is a 5-choice QA dataset testing general commonsense reasoning about the concepts from ConceptNet (Speer et al., 2017). (2) StrategyQA (Geva et al., 2021) is a binary (yes/no) QA dataset that requires models to infer the reasoning strategy. (3) OpenBookQA (Mihaylov et al., 2018) is a 4-choice QA dataset that requests reasoning based on open book as well as broad commonsense knowledge. (4) QASC (Khot et al., 2020) is an 8-choice QA dataset that requires a system to answer a question with a valid composition of basic facts using common sense. Since the gold labels for the testing sets of these datasets are not publicly available, we treat the official development set as our test set, and separate the training data into our own training set and development set.

**Evaluation Metrics** To evaluate the reasoning model's *task performance*, we use the accuracy metric and consider both ID and OOD test sets in our experiments. ID/OOD test sets are taken from the same/different dataset as the training set. To evaluate the *faithfulness* of the generated rationale to the reasoning model's predicted label, we adopt the LAS metric (Hase et al., 2020). LAS measures rationale-label consistency as how well the rationale helps a simulator model predict the reasoning model's predicted label. Following Hase et al. (2020), we implement the simulator as a fine-tuned T5-Base LM (Raffel et al., 2020a). To aggregate accuracy and LAS as a single metric, we use Normalized Relative Gain (NRG) metric (Chan et al., 2022). Across all compared methods, NRG first normalizes each of the two constituent metrics' scores as values in $[0, 1]$, then obtains the aggregate score by taking the mean of the two normalized scores.

**Implementation Details** For the rationalizing module, we use GPT-neox (Black et al., 2022), a pretrained, autoregressive LM with 20B parameters. We manually annotate 7 examples to set up the prompt for each task dataset. For the reasoning module, we adopt T5-base (Raffel et al., 2020a) with only 220 million parameters, which is around two orders of magnitude smaller than the rationalizing module. During fine-tuning, the standard training loss (Eq. 1) and our counterfactual training loss (Eq. 2) are directly combined as the overall training loss. For perturbing rationales, we randomly choose the token masking or token replacement strategy with a equal chance in each training batch. The replacing rate for token replacement is empirically set to $30\%$. We run all the experiments on the compared methods 4 times using a fixed set of random seeds and report the average results.

**Baselines** (1) *Without Rationales* is a T5-based model fine-tuned on the task dataset without using any rationales as additional input. (2) *Prompted Self-Rationalization* is a GPT-NeoX LM that learns from a few examples in the prompt to firstly generate a few short sentences as the rationale and then predict the answer. Here, we use the chain-of-thought prompting configuration from Wei et al. (2022b). (3) Distilled Self-Rationalization is a small LM (T5-base) trained on the rationales generated by the Prompted Self-Rationalization model. We implement two variants of the distillation model: a) Rationalize-First, which firstly generates the rationale and then predicts the answer, and b) Predict-First, which firstly predicts the answer and then generates the rationale. (4) NILE (Kumar & Talukdar, 2020) trains a rationalization module by fine-tuning a T5-3B model (Raffel et al., 2020a) with the rationales annotated by humans, then trains a reasoning module by fine-tuning a T5-Base model with the task dataset as in our method. We only apply NILE on the CSQA and StrategyQA datasets, since they provide human-annotated gold rationales. (5) *Standard Training* uses the same rationalize-then-reason pipeline as our method, except the reasoning module is not fine-tuned with

Table 2: **ID Results.** Task performance (accuracy), faithfulness (LAS), and Normalized Relative Gain (NRG) of the compared methods on the testing datasets. The reasoning module for the fine-tuning methods is T5-Base. We bold the results that outperform the second-best method with statistical significance ($p < 0.05$).

| Method | CSQA | | | StrategyQA | | | OBQA | | | QASC | | |
|---|---|---|---|---|---|---|---|---|---|---|---|---|
| | Acc.↑ | LAS↑ | NRG↑ | Acc.↑ | LAS↑ | NRG↑ | Acc.↑ | LAS↑ | NRG↑ | Acc.↑ | LAS↑ | NRG↑ |
| w/o Rationales | 58.68 | - | - | 58.12 | - | - | 55.85 | - | - | 35.58 | - | - |
| **Self-Rationalization** | | | | | | | | | | | | |
| Prompted GPT-neox | 38.41 | 11.66 | 0.23 | 55.31 | 1.09 | 0.47 | 33.80 | 14.67 | 0.18 | 32.61 | 32.01 | 0.33 |
| Prompted GPT-3 | **73.50** | 1.38 | 0.50 | **66.53** | 0.60 | 0.77 | - | - | - | - | - | - |
| Distill. Explain-First | 51.97 | 11.30 | 0.41 | 50.20 | 1.29 | 0.33 | 48.90 | 13.76 | 0.41 | 33.34 | 31.82 | 0.40 |
| Distill. Predict-First | 55.77 | 6.86 | 0.37 | 54.61 | -2.68 | 0.13 | 50.25 | 12.30 | 0.33 | 34.53 | 18.48 | 0.18 |
| **Pipeline** | | | | | | | | | | | | |
| NILE | 57.60 | 18.23 | 0.64 | 57.31 | 2.17 | 0.62 | - | - | - | - | - | - |
| Standard Training | 59.48 | 18.75 | 0.68 | 57.11 | 1.50 | 0.56 | 56.65 | 17.03 | 0.82 | 37.50 | 37.91 | 0.94 |
| Dropout Context | 59.64 | 20.40 | 0.72 | 51.45 | 0.62 | 0.31 | 57.55 | **18.76** | 0.97 | 35.37 | 37.54 | 0.73 |
| PINTO | 61.67 | **24.22** | **0.83** | 60.87 | **3.35** | **0.81** | **58.85** | 18.02 | 0.94 | 37.82 | **38.98** | **1.00** |
| - Masking Only | 60.46 | 17.44 | 0.67 | 59.12 | 1.74 | 0.64 | 58.35 | 13.06 | 0.55 | 37.39 | 34.06 | 0.84 |
| - Replacement Only | 60.38 | 22.54 | 0.78 | 58.72 | 2.11 | 0.66 | 58.10 | 18.01 | 0.93 | 37.47 | 34.61 | 0.86 |

Table 3: **OOD Results.** Performance (accuracy) of the compared methods, which are firstly trained on a source dataset and then directly predict on a target dataset (denoted as $source \rightarrow target$).

| Method | CSQA→OBQA | CSQA→QASC | OBQA→CSQA | QASC→CSQA | QASC→OBQA |
|---|---|---|---|---|---|
| w/o Rationales | 32.05 | 39.17 | 24.87 | 45.74 | 34.90 |
| Distill. Explain-First | 24.85 | 31.43 | 23.05 | 43.16 | 31.55 |
| Distill. Predict-First | 25.10 | 32.26 | 26.43 | 45.17 | 30.50 |
| NILE | 32.40 | 40.93 | - | - | - |
| Standard Training | 31.05 | 40.04 | 25.37 | 47.71 | 34.50 |
| Dropout Context | 32.30 | 38.85 | 23.01 | 44.27 | 32.90 |
| PINTO | **34.90** | **42.25** | **27.66** | **48.03** | **35.75** |

the counterfactual training loss. (6) *Dropout Context* is the same as the Standard Training baseline, except the question is randomly dropped out from the input while fine-tuning the reasoning module. This is a strategy used in prior work to encourage the reasoning module to make good use of the input rationales (Hase et al., 2020).

Further, we also consider two variants of PINTO, namely *Token Masking Only* and *Token Replacement Only* as baselines. These baselines only adopt token masking or token replacement for perturbing rationale tokens, respectively.

## 5 EXPERIMENTS

### 5.1 MAIN RESULTS

**In-Distribution (ID) Performance** We first evaluate all methods on ID test sets. Table 2 shows the task performance of these methods, with fine-tuning methods using T5-Base as the reasoning module. We have the following two observations. First, the Prompted Self-Rationalization baseline (using the 20B-parameter GPT-NeoX) generally does not outperform the fine-tuning methods while the GPT-3 version is reported to achieve 73.50 and 66.53 in accuracy on CSQA and StrategyQA, respectively (Wei et al., 2022b). This validates that Prompted Self-Rationalization requires very large LMs to work effectively (Wei et al., 2022a). Second, simply augmenting the reasoning module with rationales (as in Standard Training) does not always lead to better results compared with the Without Rationales baseline since the rationales may not be properly utilized. The Dropout Context baseline helps to address this issue in some, but not all cases, while PINTO consistently yields the best accuracy in most of the cases. We have similar observations from results using RoBERTa-Large as the reasoning module (Table 5 of §A.1). This demonstrates the effectiveness of our counterfactual regularization method in improving ID generalization.

**Out-of-Distribution (OOD) Performance** To further demonstrate the generalizability brought by faithful reasoning over rationales, we also investigate the performance of our method on OOD test

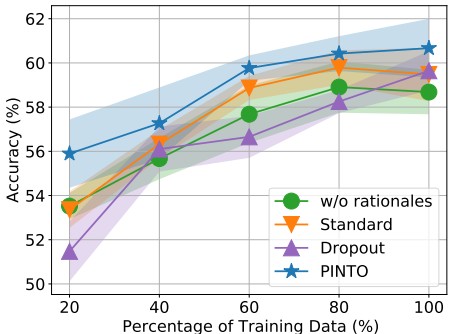

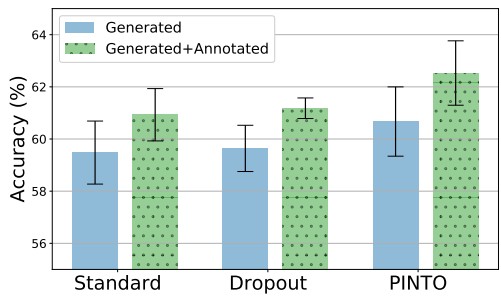

Figure 4: **Low-Resource Learning.** Performance (accuracy) of different fine-tuned models in low-resource settings on CSQA.

Figure 5: **Rationale Quality Analysis.** Accuracy of models with both generated and annotated rationales vs. models using only generated rationales on CSQA.

sets. The intuition is that by utilizing rationales faithfully rather than fitting only the ID training data, our model achieves better OOD generalization without any fine-tuning. Table 3 shows the OOD performance of all the fine-tuning methods using T5-Base. We conclude that rationales are helpful in improving the generalizability of the model to a dataset unseen during fine-tuning. Among all the methods utilizing rationales, our method yields the best OOD performance, which confirms the benefit of faithful reasoning. A consistent conclusion can be made from the results based on RoBERTa-Large (Table 6 of §A.1).

**Rationale-Label Association** Table 2 also reports the faithfulness of all the methods involving rationalization measured by LAS. We observe that PINTO achieves a much higher score compared with the baselines except on OpenBookQA. This demonstrates that counterfactual regularization helps the reasoning module make predictions more faithfully with respect to the rationales.

## 5.2 PERFORMANCE ANALYSIS

**How do different perturbation strategies contribute to the overall performance?** Table 2 shows the results of the ablation study where we only conduct Token Masking or Token Replacement when perturbing the rationale tokens. From more cases, we note that Token Replacement leads to both better accuracy and faithfulness compared with Token Masking. This is because Token Replacement perturbs the semantics of the rationales more severely, thus further forcing the reasoning module to properly make use of the rationales. Our method yields the best results when both types of perturbation are conducted, which validates that these two strategies consider comprehensively the different ways in which a reasoning module could use the rationales improperly.

**Can faithful rationales lead to better low-resource performance?** We also investigate whether, with counterfactual training, the reasoning module can be fine-tuned with less training data. Figure 4 shows the accuracy of all the fine-tuning methods. We can observe that our method consistently outperforms the baselines at different percentages of training data. The observed larger performance gap is larger when less training data is used, demonstrating the data efficiency of our method.

**Can we refine the reasoning behavior via rationales?** One important application of faithful reasoning is that rationales provide a way to refine the behavior of a model, i.e., we can correct reasoning mistakes by providing a better rationale. To verify this, we make use of ECQA (Aggarwal et al., 2021) which augments CSQA with human-annotated rationales. We directly provide the human-annotated rationales to the fine-tuned reasoning modules to obtain its oracle results, shown in Figure 5. We see that human-annotated rationales generally lead to performance gain for all fine-tuning methods whereof the gain of our method is the largest. This again showcases the merits of ensuring the faithful reasoning on rationales in refining a system.

**Is our method more sensitive to perturbed rationales?** Intuitively, higher rationale faithfulness (i.e., stronger connection between the rationale the and reasoning module's behavior) should yield greater sensitivity to noisily perturbed rationales. In other words, higher performance drop (sensitivity) signals higher faithfulness. To verify this, we conduct a stress test. We choose CSQA and OpenBookQA and replace each question in the testing set with a randomly sampled question but

still keep the original answer choices. We then prompt our rationalizing module with the replaced question and the original choices to obtain a set of perturbed rationales. We finally provide the perturbed rationales to the reasoning module. Our results in Table 4 show that PINTO achieves a significantly higher performance drop than the other two methods (esp. on OBQA), indicating that counterfactual regularization is effective in improving rationale faithfulness.

## 6 RELATED WORK

Extensive work has been done on solving implicit reasoning tasks by augmenting reasoning LMs with external knowledge beyond the task input. Prior works have explored retrieving implicit knowledge from: (1) knowledge graphs (Lin et al., 2019; Feng et al., 2020; Wang et al., 2020; Yan et al., 2021; Chan et al., 2021; Raman et al., 2021), (2) web corpora (Lv et al., 2020; Chen et al., 2017; Yang et al., 2015; Ryu et al., 2014), or (3) pretrained LMs (Shwartz et al., 2020; Liu et al., 2021; Bosselut et al., 2019; Shin et al., 2020). Although knowledge retrieval has shown to be helpful in boosting reasoning LMs' task performance, it may not necessarily explain

Table 4: **Sensitivity to Noisy Rationales.** We use perturbed rationales during inference as a stress test and report the performance drop of the compared methods. .

| Model | CSQA | OBQA |
|---|---|---|
| Standard Training | 0.88 | 0.35 |
| Dropout Context | 2.06 | 0.55 |
| PINTO | **2.62** | **1.55** |

the decisions made by the LM. Given the lack of transparency in neural LMs' complex behavior (Rudin, 2019; Caruana, 2019), model explainability is important for promoting human trust in NLP systems for high-stakes decision-making (Doshi-Velez & Kim, 2017; Lipton, 2018; Bender et al., 2021). We focus on rationale generation in this work as a way to both improve an LM's task performance and provide justification for its predictions.

Prior works on free-text rationale generation can be grouped into three paradigms. In the *fine-tuned self-rationalizing* paradigm, a single LM is fine-tuned to jointly generate the task output and rationale (Narang et al., 2020; Marasović et al., 2022; Zelikman et al., 2022; Li et al., 2022). Since the LM parameters are shared across two relatively dissimilar objectives, they often perform worse than non-rationalizing LMs (Wiegreffe et al., 2020; Narang et al., 2020). Notably, this paradigm requires expensive rationale annotations for all training instances. In the *prompted self-rationalizing* paradigm, a single LM is instead frozen and prompted to jointly generate the task output and rationale, with the prompt consisting of a few input-output-rationale demonstrations (Wei et al., 2022b). This paradigm performs well and only needs a few rationale annotations for the prompt, but it is computationally prohibitive since it generally requires very large-scale LMs to work effectively (Lampinen et al., 2022; Wei et al., 2022b). In the *pipeline-rationalizing* paradigm, a fine-tuned rationalizing LM first generates the rationale, which is then used as input for a separate fine-tuned reasoning LM to generate the output (Kumar & Talukdar, 2020; Rajani et al., 2019). Here, the generated rationale forms a discrete (*i.e.,* non-differentiable) bottleneck between the two modules, which complicates end-to-end training and can hurt task performance (Wiegreffe et al., 2020; Hase et al., 2020). Additionally, the dedicated rationalizing LM requires extra rationale annotation/computation costs. Moreover, none of these paradigms has a mechanism for regularizing the rationale generation to *faithfully* reflect the reasoning process of the LM, without hurting task performance. PINTO avoids these limitations by rationalizing via prompt-based learning (using a frozen medium-scale LM), then reasoning over the task input and rationale via counterfactual regularization (using a fine-tuned small-scale LM).

## 7 CONCLUSION

This paper presents PINTO, an LM pipeline that rationalizes with prompt-based learning and reasons via counterfactual regularization. Through prompting, we remove the need for expensive human annotation and leverage the massive knowledge encoded in a medium-sized LM to perform rationalization. With counterfactual regularization in addition to standard training objective, our reasoning module learns to reason over the generated rationales more faithfully. Experiments show that our method outperforms baselines on both in-distribution and out-of-distribution datasets in accuracy, while providing higher faithfulness. Our analysis also shows that we can further improve task performance with a more faithful reasoning module and refined rationales.

## ACKNOWLEDGEMENT

We thank the anonymous reviewers and all the collaborators in USC INK research lab for their valuable feedback. This material is based upon work sponsored by the DARPA MCS program under Contract No. N660011924033 with the United States Office Of Naval Research.

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

# A APPENDIX

## A.1 EXPERIMENTS WITH ROBERTA AS THE REASONING MODULE

Table 5-6 show both the ID and OOD results based on RoBERTa-Large as the reasoning module. The observations are consistent with Table 2-3 where we fine-tune T5-Base as the reasoning module.

## A.2 ABLATION ON THE LM SIZE FOR THE RATIONALIZING MODULE

Table 7 shows the results of the Pipeline approaches using LMs with different model sizes as the rationalizing module.

## A.3 VARIANCE STATISTICS OF ALL THE FINE-TUNED MODELS

Table 8 and Table 9 show the variance statistics (standard deviation) along with ID and OOD task performance (accuracy) of the fine-tuned methods from Table 2 and Table 3.

Table 5: **ID Results.** Task performance (accuracy), faithfulness (LAS), and Normalized Relative Gain (NRG) of the compared methods on the testing datasets. The rationalizing module is GPT-neox (20B) while the reasoning module for the fine-tuning methods is RoBERTa-Large. We bold the results that outperform the second-best method with statistical significance ($p < 0.05$).

| Method | CSQA | | | StrategyQA | | | OBQA | | | QASC | | |
|---|---|---|---|---|---|---|---|---|---|---|---|---|
| | Acc.↑ | LAS↑ | NRG↑ | Acc.↑ | LAS↑ | NRG↑ | Acc.↑ | LAS↑ | NRG↑ | Acc.↑ | LAS↑ | NRG↑ |
| w/o Rationales | 71.44 | - | - | 53.76 | - | - | 66.40 | - | - | 53.32 | - | - |
| NILE | 74.32 | 28.81 | 0.43 | 62.85 | 0.69 | 0.42 | - | - | - | - | - | - |
| Standard Training | 73.98 | 29.50 | 0.63 | 62.78 | 2.29 | 0.68 | 69.9 | 26.04 | 0.46 | 53.64 | 50.76 | 0.03 |
| Dropout Context | 71.81 | **30.13** | 0.50 | 60.27 | 2.58 | 0.32 | 68.6 | 25.06 | 0.00 | 53.48 | 51.55 | 0.50 |
| PINTO | **74.73** | 29.36 | **0.71** | **63.33** | **3.66** | **1.00** | **72.6** | **26.72** | **1.00** | **56.32** | 51.19 | **0.77** |

Table 6: **OOD Results.** Performance (accuracy) of the compared methods using RoBERTa-Large as the reasoning module, which are firstly fine-tuned on a source dataset and then directly predict on a target dataset (denoted as $source \rightarrow target$).

| Method | CSQA→OBQA | CSQA→QASC | OBQA→CSQA | QASC→CSQA | QASC→OBQA |
|---|---|---|---|---|---|
| w/o Rationales | 49.55 | 53.67 | 44.51 | 60.48 | 54.15 |
| NILE | 42.85 | 50.16 | - | - | - |
| Standard Training | 54.60 | 55.75 | 46.62 | 60.26 | 55.95 |
| Dropout Context | 52.55 | 55.40 | 41.01 | 58.60 | 58.90 |
| PINTO | 54.30 | **56.83** | **46.95** | **61.61** | **60.95** |

Table 7: **Ablation on the LM with different model sizes as the rationalizing module.** We bold the results that outperform the second-best method with statistical significance ($p < 0.05$).

| Method | GPT-2 (1.5B) | | GPT-J (6B) | | GPT-neox (20B) | | GPT-3 (175B) | |
|---|---|---|---|---|---|---|---|---|
| | Acc.↑ | LAS↑ | Acc.↑ | LAS↑ | Acc.↑ | LAS↑ | Acc.↑ | LAS↑ |
| Standard Training | 59.19 | 13.64 | 59.44 | 15.81 | 59.48 | 18.75 | 59.40 | 22.99 |
| Dropout Context | 58.82 | 13.33 | 58.48 | 15.70 | 59.64 | 20.40 | 58.64 | 23.51 |
| PINTO | **61.10** | **14.63** | **60.73** | **16.09** | **61.67** | **24.22** | **61.32** | **23.54** |

Table 8: **ID Results with variance statistics**. Task performance (accuracy) and variance statistics (standard deviation) of the fine-tuned methods from Table 2.

| Method | CSQA | | StrategyQA | | OpenBookQA | | QASC | |
|---|---|---|---|---|---|---|---|---|
| | Acc.↑ | STD | Acc.↑ | STD | Acc.↑ | STD | Acc.↑ | STD |
| w/o Rationales | 58.68 | 0.19 | 58.12 | 2.53 | 55.85 | 0.90 | 35.58 | 0.64 |
| Distill. Self-Ra. (Explain-First) | 51.97 | 0.41 | 50.20 | 4.03 | 48.90 | 0.78 | 33.34 | 1.18 |
| Distill. Self-Ra. (Predict-First) | 55.77 | 0.69 | 54.61 | 1.87 | 50.25 | 3.45 | 34.53 | 0.70 |
| Standard Training | 59.48 | 0.41 | 57.11 | 2.66 | 56.65 | 0.70 | 37.50 | 0.55 |
| Dropout Context | 59.64 | 0.89 | 51.45 | 2.76 | 57.55 | 1.08 | 35.37 | 1.87 |
| PINTO | 61.67 | 0.31 | 60.87 | 2.66 | 58.85 | 1.37 | 37.82 | 1.48 |

## A.4 HUMAN EVALUATION ON THE RATIONALES

We conducted a human evaluation of 100 generated rationales from the CSQA dataset. The evaluation is a head-to-head comparison between the human-annotated rationales and the machine-generated rationales. Annotators were asked to judge for 5 dimensions on a 3-point Likert scale following a prior work (Wiegreffe et al., 2021): 1) Factuality (How factual is this Explanation?) 2) Grammaticality (Is this Explanation grammatical?) 3) New Info (Does the Explanation provide new

Table 9: **OOD Results with variance statistics**. Task performance (accuracy) and variance statistics (standard deviation) of the fine-tuned methods from Table 3.

| Method | CS→OB Acc.↑ | STD | CS→QASC Acc.↑ | STD | OB→CS Acc.↑ | STD | QASC→CS Acc.↑ | STD | QASC→OB Acc.↑ | STD |
|---|---|---|---|---|---|---|---|---|---|---|
| w/o Rationales | 32.05 | 0.90 | 39.17 | 1.54 | 24.87 | 1.52 | 45.74 | 1.19 | 34.90 | 2.06 |
| Distill. Self-Ra. (Explain-First) | 24.85 | 1.51 | 31.43 | 0.85 | 23.05 | 1.42 | 43.16 | 1.00 | 31.55 | 0.71 |
| Distill. Self-Ra. (Predict-First) | 25.10 | 1.40 | 32.26 | 0.64 | 26.43 | 1.42 | 45.17 | 0.88 | 30.50 | 0.87 |
| Standard Training | 31.05 | 1.10 | 40.04 | 1.13 | 25.37 | 0.7 | 47.71 | 1.15 | 34.50 | 1.46 |
| Dropout Context | 32.30 | 1.57 | 38.85 | 0.70 | 23.01 | 1.14 | 44.27 | 2.36 | 32.90 | 2.06 |
| PINTO | 34.90 | 1.37 | 42.25 | 0.35 | 27.66 | 1.86 | 48.03 | 0.97 | 35.75 | 1.47 |

Table 10: **Human evaluation on the human-annotated and machine-generated rationales for CSQA.** A fair level of agreement measured by Fleiss Kappa (k=0.34) is obtained among the annotators.

| Rationale Source | Factuality | Grammaticality | New Info | Supports Answer | Completeness |
|---|---|---|---|---|---|
| Human | 0.94 | 0.98 | 0.71 | 0.90 | 0.82 |
| Generated | 0.91 | 0.99 | 0.69 | 0.87 | 0.65 |

Table 11: **Case Study.** Generated rationales and model predictions from CSQA and StrategyQA.

| Questions and Choice-specific Rationales | Standard | PINTO |
|---|---|---|
| Q: If you have leftover cake, where would you put it? 
 (a) quandry: *Quandry is a place where you put things that you don't know ...* 
 (b) refrigerator: *Refrigerator is used to store food.* 
 (c) oven: *Oven is used to bake cakes.* | oven ✗ | refrigerator ✓ |
| Q: Has every astronaut survived their space journey? 
 yes: *Astronauts have died in space. For example, the Challenger astronauts died in the Challenger space shuttle explosion.* 
 no: *The first astronaut to die was Yuri Gagarin. Gagarin died in a plane crash.* | yes ✗ | no ✓ |

facts, information, or reasoning not stated in the Question and Answer?) 4) Supports Answer (Is the Explanation relevant to justifying the Answer?) 5) Completeness (Does the Explanation provide enough information to jusify the answer?)

We obtain a fair level of agreement measured by Fleiss Kappa (k=0.34) for the evaluation. The results in Table 10 show that machine-generated results are competitive with human annotation on most of the evaluating dimensions. Generated rationales are even judged to be more grammatical than human annotations. As for completeness, generated rationales are slightly worse than human annotations. We think this is because the human annotators were explicitly encouraged to provide more comprehensive rationales when annotating the CSQA dataset (Aggarwal et al., 2021).

## A.5 CASE STUDY

We provide concrete examples in Table 11 to showcase how our prompted LM rationalizes for correct and incorrect choices and how PINTO reasons more faithfully compared with the Standard baseline. In the question (second row) from CSQA, we can see that for incorrect choices, the generated rationales do not support them to be the answer while the one for the correct choice *refrigerator* does. In the question (third row) from StrategyQA, the rationale for the correct choice *yes* is sound and reasonable while the rationale for the incorrect choice *no* is factually correct but does not answer the question directly (*died in a plane crash* vs. *died in the space journey*). For both questions, PINTO properly leverages the rationales and make the correct predictions while the Standard baseline fails.

## A.6 Prompts for Rationalization

Table 12- 15 show the complete prompts we use to obtain rationales from LM for CSQA, StrategyQA, OpenBookQA and QASC datasets.

Table 12: The complete prompt of rationalization for CommonsenseQA. We adopt the rationalizations annotated in Aggarwal et al. (2021).

---

**Q**: What do people use to absorb extra ink from a fountain pen?
**Answer Choices**: (a) shirt pocket (b) calligrapher's hand (c) inkwell (d) desk drawer (e) blotter
**A**: The answer is blotter. *Blotting paper absorbs liquids like ink well.*

**Q**: What home entertainment equipment requires cable?
**Answer Choices**: (a) radio shack (b) substation (c) cabinet (d) television (e) desk
**A**: The answer is television. *Cable can be fixed to a television. Television is a home entertainment equipment.*

**Q**: The fox walked from the city into the forest, what was it looking for?
**Answer Choices**: (a) pretty flowers (b) hen house (c) natural habitat (d) storybook (e) dense forest
**A**: The answer is natural habitat. *Forests are one of the main natural habitats of foxes.*

**Q**: Sammy wanted to go to where the people were. Where might he go?
**Answer Choices**: (a) populated areas (b) race track (c) desert (d) apartment (e) roadblock
**A**: The answer is populated areas. *Populated areas are where there are a lot of people.*

**Q**: Where do you put your grapes just before checking out?
**Answer Choices**: (a) mouth (b) grocery cart (c) super market (d) fruit basket (e) fruit market
**A**: The answer is grocery cart. *Grocery cart is used in stores by customers to collect purchases. Checking out of purchases is done in checkout area of stores.*

**Q**: Google Maps and other highway and street GPS services have replaced what?
**Answer Choices**: (a) united states (b) mexico (c) countryside (d) atlas (e) oceans
**A**: The answer is atlas. *Atlas are replaced by more precise Google maps, other highway and street GPS services. One can get much more precise data with the help of Google maps and Street GPS services.*

**Q**: Before getting a divorce, what did the wife feel who was doing all the work?
**Answer Choices**: (a) harder (b) anguish (c) bitterness (d) tears (e) sadness
**A**: The answer is bitterness. *Bitterness is the resentful feeling of anger at being treated unfairly. Doing all the work means being treated unfairly.*

---

Table 13: The complete prompt of rationalization for StrategyQA. The prompt is adapted from Wei et al. (2022b).

---

**Q**: Do hamsters provide food for any animals?
**A**: The answer is yes. *Hamsters are prey animals. Prey animals provide food for predators.*

**Q**: Could Brooke Shields succeed at University of Pennsylvania?
**A**: The answer is yes. *Brooke Shields went to Princeton University. Princeton University is about as academically rigorous as the University of Pennsylvania.*

**Q**: Hydrogen's atomic number squared exceeds number of Spice Girls?
**A**: The answer is no. *Hydrogen has an atomic number of 1. 1 squared is 1. There are 5 Spice Girls.*

**Q**: Is it common to see frost during some college commencements?
**A**: The answer is yes. *College commencement ceremonies can happen in December, May, and June. December is in the winter, so there can be frost.*

**Q**: Could a llama birth twice during War in Vietnam (1945-46)?
**A**: The answer is no. *The War in Vietnam was 6 months. The gestation period for a llama is 11 months, which is more than 6 months.*

**Q**: Would a pear sink in water?
**A**: The answer is no. *The density of a pear is about 0.6 $g/cm^3$, which is less than water. Objects less dense than water float.*

---

Table 14: The complete prompt of rationalization for OpenBookQA. We use the basic facts provided by Mihaylov et al. (2018) as the rationalization.

---

**Q**: The sun is responsible for
**Answer Choices**: (a) puppies learning new tricks (b) children growing up and getting old
(c) flowers wilting in a vase (d) plants sprouting, blooming and wilting
**A**: The answer is plants sprouting, blooming and wilting. *A plant requires sunlight for photosynthesis, which accumulates resources required for sprouting, blooming and wilting.*

**Q**: When standing miles away from Mount Rushmore
**Answer Choices**: (a) the mountains seem very close (b) the mountains are boring
(c) the mountains look the same as from up close (d) the mountains seem smaller than in photographs
**A**: The answer is the mountains seem smaller than in photographs.*When an object is far away, it takes up less of your field of view, and so seems smaller than in the photographs.*

**Q**: When food is reduced in the stomach
**Answer Choices**: (a) the mind needs time to digest (b) take a second to digest what I said
(c) nutrients are being deconstructed (d) reader's digest is a body of works
**A**: The answer is nutrients are being deconstructed. *The stomach is part of the digestive system. The breaking down of food into nutrients occurs in the digestive system.*

**Q**: Poison causes harm to which of the following?
**Answer Choices**: (a) a Tree (b) a robot (c) a house (d) a car
**A**: The answer is a Tree.*A tree is a living thing. Poison causes harm to living things.*

**Q**: A magnet will stick to
**Answer Choices**: (a) a belt buckle (b) a wooden table
(c) a plastic cup (d) a paper plate
**A**: The answer is a belt buckle. *A belt buckle is made of metal. If a magnet is attracted to a metal then that magnet will stick to that metal.*

**Q**: Deer are less safe in the woods because wolves
**Answer Choices**: (a) have fur (b) howl (c) have claws (d) have tails
**A**: The answer is have claws. *Claws are used by wolves to catch prey like deer.*

**Q**: An electric car causes
**Answer Choices**: (a) more CO2 emissions (b) equal CO2 emissions (c) electric emissions
(d) less CO2 emissions
**A**: The answer is less CO2 emissions. *An electric car uses less gasoline than a regular car and thus causes less CO2 emissions.*

---

Table 15: The complete prompt of rationalization for QASC. We adapt the supporting facts from Khot et al. (2020) as the rationalization.

---

**Q**: How do you reduce pollution?
**Answer choices**: (a) igniting fuel and oxidiser (b) transportation technology (c) wasting (d) not recycling
(e) burning fossil fuels (f) converting electricity to heat (g) water conservation (h) using less resources
**A**: The answer is using less resources. *Conserving resources has a positive impact on the environment. Use of resources affects the environment such as pollution.*

**Q**: what will move to another area if their habitat will no longer support them?
**Answer choices**: (a) density (b) Birds (c) squids (d) humans (e) clouds (f) gravity (g) cows (h) Whales
**A**: The answer is cows. *If a habitat can no longer support animals then those animals will move to another area. Cows are social animals.*

**Q**: With the exception of allergies, what may cause a person to seek medical attention?
**Answer choices**: (a) Contact with latex (b) a tree falling (c) Organs within the body. (d) Contact with baby chicks (e) prolactin release (f) Contact with peanut butter (g) hypothyroidism (h) Contact with microorganisms
**A**: The answer is Contact with microorganisms. *Microorganisms can cause infections. Infections usually require medical treatment.*

**Q**: Lavender can induce
**Answer choices**: (a) healing (b) energy (c) hormones (d) mutations (e) Heart rate (f) growth
(g) symptoms (h) warmth
**A**: The answer is healing. *Healing requires rest. Lavender induces restful sleep.*

**Q**: what state is a liquid in when frozen?
**Answer choices**: (a) vapor (b) dense (c) gas (d) cooled (e) steam (f) solid (g) boiling (h) cold
**A**: The answer is solid. *Freezing means changing from a liquid into a solid by reducing heat energy. Liquids freeze when they change to the solid state.*

**Q**: what unites to form a diploid zygote?
**Answer choices**: (a) plant reproduction (b) Most plants (c) orchids (d) sperm and ova (e) salt and pepper
(f) predator and prey (g) honeybees (h) diploids and zygotes
**A**: The answer is sperm and ova. *Gametes then unite in fertilization and form a diploid zygote. Collectively, the sperm and the ova are also referred to as gametes .*

**Q**: What absorbs all visible light?
**Answer choices**: (a) apples (b) coal (c) Green (d) coral (e) skin (f) bamboo (g) glass (h) eyes
**A**: The answer is coal. *If an object is black then that object absorbs all visible light. Light grains are quartz, Black grains are coal.*

---

