# OpenReview forum: "PINTO: Faithful Language Reasoning Using Prompt-Generated Rationales"
_ICLR.cc/2023/Conference — ICLR 2023 poster_

### Official Review · Reviewer_WxQk · 2022-10-20

**Confidence:** 4
**Correctness:** 4
**Technical Novelty And Significance:** 3
**Empirical Novelty And Significance:** 3
**Recommendation:** 6

**Clarity, Quality, Novelty And Reproducibility:**

* Novelty: see bullet 1 above
* Quality: the paper is well written and the the experiments are thorough.
* Reproducibility: the code is provided

**Strength And Weaknesses:**

* The main strength of the paper is paying attention to the computational budget, which is one motivation behind the algorithm. I believe it will become an important direction as the pretrained models are getting larger and larger.

* The algorithm design is simple. On the other hand, the non-trivial part is the potential shortcut issue, and the authors propose a counterfactual regularization to resolve it.

### Questions

* Although I understand one argument of the paper is computation efficiency, I'm still curious about what is the performance of the fine-tuned LLM on the benchmarks. The used LLMs are still "relatively" small compared with GPT-3 and PaLM. It would be nice to have these numbers to serve an upper bound.

* For the OOD results, the proposed algorithm leverage the rational from the LLM. However, it has been demonstrated that LLM has strong zero-shot performance as well as great generalization capability.  Therefore, regardless the training data it used, when testing on the different dataset, as long as the rational provided by LLM is faithful, isn't it expected that any model built on top of it can generalize well?  In this case, is it fair to argue the proposed model has good generalization capability? or it's  actually  benefitted by LLM's good generalization capability?

* Question about robustness of the perturbed data.  When the rational is wrong, the model can still make right decision. Although the authors try to argue it's not a contradictory results, it does sound contradictory to me that, if the model is encouraged to take the rational into account when making decision, how can it not be misled if we feed a wrong rational? if it can be robust, isn't it not leveraging rationals when making decisions?  Could the author comment more on it?

* The rational qualities are also known sensitive to the prompt design. How does it affect the final algorithm performance?

**Summary Of The Paper:**

The authors propose an algorithm to leverage LLM's rational through prompt, to improve final downstream models' performance by taking the computation into account. Specifically, to save human efforts, the authors propose to use LLM to generate rationals. It then train a QA model based by giving the generated rationals. To alleviate the shortcut, they further propose a counterfactual regularization to encourage the model to learn to use the rational instead of learning the shortcut.

**Summary Of The Review:**

The paper focus on a computational efficient algorithm by using LLM, which seems important to me. The overall quality is good as aforementioned. There are some questions not clear to me, but overall I think it's a good paper.

---

> ### Author Response · Authors · 2022-11-19
> **Response to Reviewer WxQk (Part 1)**
>
> Thank you for your positive review and very helpful comments! Below are our responses and we have updated the draft accordingly.
>
> > 1. Although I understand one argument of the paper is computation efficiency, I'm still curious about what is the performance of the fine-tuned LLM on the benchmarks. The used LLMs are still "relatively" small compared with GPT-3 and PaLM. It would be nice to have these numbers to serve an upper bound.
>
> Thank you for the suggestion! We have added new experiments using another SOTA LM, T5-3B, as the reasoning module. The results below show that PINTO still outperforms the baselines in both task performance (Acc.) and faithfulness (LAS) with larger LMs. **Notably, PINTO achieves 82.39 in accuracy, which is higher than what GPT-3 is reported to achieve (73.50).** This demonstrates PINTO’s general effectiveness.
>
> | Method with T5-3B | CSQA || OBQA ||
> | --- | ---: | --: | --: | --: |
> | | Acc. | LAS | Acc. | LAS |
> | w/o Rationale | 80.94 | - | 73.6 | - |
> | Prompted Self-Rationalization (GPT-3) | 73.50 | 1.38 | - | - |
> | Standard | 81.70 | 28.22 | 79.1 | 22.34 |
> | Dropout | 81.35 | 26.90 | 78.2 | 22.13 |
> | PINTO | **82.39** | **29.78** | **80.35** | 23.16 |
>
> > 2. For the OOD results, the proposed algorithm leverage the rational from the LLM. However, it has been demonstrated that LLM has strong zero-shot performance as well as great generalization capability. Therefore, regardless the training data it used, when testing on the different dataset, as long as the rational provided by LLM is faithful, isn't it expected that any model built on top of it can generalize well? In this case, is it fair to argue the proposed model has good generalization capability? or it's actually benefitted by LLM's good generalization capability?
>
> We would like to clarify that the rationales provided by the LLM may not be faithful, since the reasoning module may still learn to ignore the rationales and make predictions by exploiting spurious reasoning shortcuts [1]. Motivated by this issue, PINTO’s counterfactual regularization is designed to encourage the reasoning module utilize the rationales in a sounder, more generalizable manner. Our results show that counterfactual regularization works well in improving both task performance and faithfulness. In particular, our results suggest that counterfactual regularization helps keep the reasoning module from overfitting the training data and thus failing on OOD data.
>
> [1] Ruben et al “Shortcutted commonsense: Data spuriousness in deep learning of commonsense reasoning.” 2021
>
> > 3. Question about robustness of the perturbed data. When the rational is wrong, the model can still make right decision. Although the authors try to argue it's not a contradictory results, it does sound contradictory to me that, if the model is encouraged to take the rational into account when making decision, how can it not be misled if we feed a wrong rational? if it can be robust, isn't it not leveraging rationals when making decisions? Could the author comment more on it?
>
> Thank you for your keen observation! Following your feedback, we have revised our interpretation of the results. First, as shown in the table below, we now also report Original - Perturbed, which is the delta between the original performance and perturbed performance. Second, you are correct that higher rationale faithfulness (i.e., stronger connection between the rationale the and reasoning module’s behavior) should intuitively yield greater sensitivity to noisily perturbed rationales. In other words, higher delta (sensitivity) signals higher faithfulness. Third, our results actually show that PINTO achieves a significantly higher delta than the other two methods (esp. on OBQA), indicating that counterfactual regularization is effective in improving rationale faithfulness. We have updated the paper to reflect these changes.
>
> | Model / Rationale | CSQA ||| OBQA |||
> | --- | ---: | ---: | ---: | ---: | ---: | ---: |
> | | Original | Perturbed | Original - Perturbed | Original | Perturbed | Original - Perturbed |
> | Standard Training | 59.48 | 58.60 | 0.88 | 56.65 | 56.30 | 0.35 |
> | Dropout Context | 59.64 | 57.58 | 2.06 | 57.55 | 57.00 | 0.55 |
> | PINTO | **61.67** | **59.05** | **2.62** | **58.85** | **57.30** | **1.55** |

---

> > ### Author Response · Authors · 2022-11-19
> > **Response to Reviewer WxQk (Part 2)**
> >
> > > 4. The rational qualities are also known sensitive to the prompt design. How does it affect the final algorithm performance?
> >
> > Many works have explored the effects of prompt design on LM behavior, and this is still an open research area. While an extensive investigation of optimal prompt design is beyond the scope of our work, in our preliminary experiments, we found that the LMs we considered were relatively robust to different prompt designs. This finding aligns with those reported in Wei et al. 2022b [1], which empirically demonstrates that LMs are robust to different annotators, exemplars, and the order and number of the exemplars of the prompt. As further evidence of generated rationales’ quality, we also refer the reviewer to Wiegreffe et al. 2021 [2], which shows that the rationales generated by LMs are actually preferable over human-annotated rationales.
> >
> > [1] Wei et al. “Chain of thought prompting elicits reasoning in large language models.” 2022.
> >
> > [2] Wiegreffe et al. “Reframing human-ai collaboration for generating free-text explanations.” 2021.

---

> ### Author Response · Authors · 2022-11-24
> **Friendly reminder to respond to author rebuttal**
>
> Dear Reviewer WxQk,
>
> Thank you again for your review! We are happy to hear that you appreciated PINTO’s novelty, performance/efficiency improvements, experiments, motivation, and presentation.
>
> Based on your thoughtful feedback, we wrote a detailed rebuttal covering the following points:
> - Comparison between PINTO and SOTA methods
> - Connection between PINTO’s rationalization faithfulness and its OOD generalization ability
> - Connection between PINTO’s rationalization faithfulness and its robustness to perturbed rationales
> - Effects of prompt design on PINTO’s generated rationale quality and performance
>
> We would love to hear your thoughts about our rebuttal, including whether it sufficiently addresses your concerns and questions. If you believe that our rebuttal is satisfactory, it would be great if you could consider increasing your score. Any feedback is welcome and greatly appreciated!
>
> Sincerely,
>
> Paper5818 Authors

---

> ### Author Response · Authors · 2022-11-30
> **Friendly follow-up reminder to respond to author rebuttal**
>
> Dear Reviewer WxQk,
>
> Just wanted to follow up on our previous reminder message!
>
> We would love to hear your thoughts about our rebuttal, including whether it sufficiently addresses your concerns and questions. If you believe that our rebuttal is satisfactory, it would be great if you could consider increasing your score. Any feedback is welcome and greatly appreciated!
>
> Sincerely,
>
> Paper5818 Authors

---

> ### Author Response · Authors · 2022-12-06
> **Another friendly follow-up reminder to respond to author rebuttal**
>
> Dear Reviewer WxQk,
>
> Just wanted to follow up on our previous reminder message, since we are quickly approaching the Discussion Stage 2 deadline on December 12!
>
> We would love to hear your thoughts about our rebuttal, including whether it sufficiently addresses your concerns and questions. If you believe that our rebuttal is satisfactory, it would be great if you could consider increasing your score. Any feedback is welcome and greatly appreciated!
>
> Sincerely,
>
> Paper5818 Authors

---

### Official Review · Reviewer_gqDU · 2022-10-24

**Confidence:** 4
**Correctness:** 4
**Technical Novelty And Significance:** 2
**Empirical Novelty And Significance:** Not applicable
**Recommendation:** 5

**Clarity, Quality, Novelty And Reproducibility:**

Clarity: the paper is not very clear on the contribution and something misses baseline in the main table (e.g. Table 2, some results are in the caption of the table). More over the name of the baselines (e.g., Standard or Dropout) are very hard to follow.
Quality: the paper has some merit.
Novelty: not very novel (see weakness).
Reproducibility: with some effort, but it's reproducible.

**Strength And Weaknesses:**

Weaknesses
It's hard to grasp the novelty and the contribution of this paper. For instance, the main message, or at least if I understood this paper correctly, is that mid size LM (20B) can be use to generate rationale (which per se is not novel) and the by fine tuning on it a smaller model the performance are "close" to simple baselines (which are not SOTA nor this is novel per se). Some detailed points:
- Prompted Self-Rationalization uses larger model to solve the task directly, however the model is very large. Then why not use those generation to train a smaller model as in [1,2] directly?
- Based on the provided analysis that  more faithful rationales can improve the performance, what is the quality of the generated rationales? is there a small scale human eval to quantify this.


[1] https://arxiv.org/abs/2110.07178
[2] https://arxiv.org/pdf/2104.08826.pdf

**Summary Of The Paper:**

The authors, presents PINTO a two stage pipeline to improve rationale-based language reasoning. The first step is to prompt a language model (20B in this case) to generate rationale given a question, while the second step is to fine tune a smaller (<1B) LM with the pair (Question, generated rationale) -> Answer.

The authors experimented PINTO on the CSR benchmarks (CommonsenseQA, StrategyQA, OpenBookQA, QASC). The results show that PINTO works better across the four dataset when compared to other baseline settings. The authors also reports interesting ablations and case studies.

**Summary Of The Review:**

The paper lack of a major contribution and major focus. After the discussion between the reviewers, I decide to keep the increase my score (post-rebuttal), to 5.

---

> ### Author Response · Authors · 2022-11-19
> **Response to Reviewer gqDU (Part 1)**
>
> Thank you for your helpful comments! Below are our responses and we have updated the draft accordingly.
>
> > 1. It's hard to grasp the novelty and the contribution of this paper. For instance, the main message, or at least if I understood this paper correctly, is that mid size LM (20B) can be use to generate rationale (which per se is not novel) and the by fine tuning on it a smaller model the performance are "close" to simple baselines (which are not SOTA nor this is novel per se)
>
> **Novelty**
>
> Sec. 1 and Fig. 1 (as well as “Summary of Contributions” in the General Response) describe the distinctions between PINTO and the paradigms used in prior works. We summarize these points below:
> - First, unlike fine-tuned self-rationalization and pipeline rationalization, PINTO does not require human-annotated rationale to train the rationalizing module.
> - Second, unlike prompted self-rationalization, PINTO does not require the rationalizing module to be extremely large (i.e., over 100B parameters).
> - Third, none of these prior works consider whether the generated rationales are faithful to the model’s reasoning process. Without any way to ensure rationale faithfulness, it is doubtful that prior works’ generated rationales are reliable for understanding the model’s decision-making, especially if the model is behaving problematically. To address this issue, PINTO’s counterfactual regularization encourages the reasoning module’s behavior to align with the generated rationale. We provide support for PINTO’s faithfulness in Sec. 5.2 and Fig. 6, where we show that the strong connection between PINTO’s rationalizing and reasoning enables it to achieve larger performance gains when its rationales are augmented with human feedback, compared to baselines without counterfactual regularization.
>
> We will update the paper to better convey PINTO’s novelty.
>
> **Comparison to SOTA Methods**
>
> In our submission, we focused on T5-Base as the reasoning module. However, PINTO can also accommodate larger reasoning module architectures, which are critical to achieving high task performance. Below, we show CSQA and OBQA results for using SOTA LM architectures (i.e., RoBERTa-Large and T5-3B) as the reasoning module. As shown in the results below, using either RoBERTa-Large or T5-3B, PINTO still outperforms the baselines in both task performance (Acc.) and faithfulness (LAS). In particular, PINTO with T5-3B yields SOTA performance on both CSQA and OBQA. This suggests that PINTO is generally effective across different LM architectures.
>
> | Method with RoBERTa-Large | CSQA || OBQA ||
> | --- | ---: | ---: | ---: | ---: |
> | | Acc. | LAS | Acc. | LAS |
> | w/o Rationale | 73.57 | - | 66.4 | - |
> | Standard | 73.98 | 29.50 | 69.9 | 26.04 |
> | Dropout | 71.81 | **​​30.13** | 68.6 | 25.06 |
> | PINTO | **74.73** | 29.36 | **72.6**| **26.72** |
>
> | Method with T5-3B | CSQA || OBQA ||
> | --- | ---: | ---: | ---: | ---: |
> | | Acc. | LAS | Acc. | LAS |
> | w/o Rationale | 80.94 | - | 73.6 | - |
> | Standard | 81.70 | 28.22 | 79.1 | 22.34 |
> | Dropout | 81.35 | 26.90 | 78.2 | 22.13 |
> | PINTO | **82.39** | **29.78** | **80.35** | **23.16** |

---

> > ### Author Response · Authors · 2022-11-19
> > **Response to Reviewer gqDU (Part 2)**
> >
> > > 2. Prompted Self-Rationalization uses larger model to solve the task directly, however the model is very large. Then why not use those generation to train a smaller model as in [1,2] directly?
> >
> > Thank you for the suggestion! In response, we have added Distilled Self-Rationalizating as a new baseline. In Distilled Self-Rationalization, a small (student) Self-Rationalizing model (T5-Base) is supervised using the output of a large (teacher) Prompted Self-Rationalization model (GPT-NeoX). Recall that a Prompted Self-Rationalization model’s input sequence consists of a demonstration prompt followed by the given question, while its output sequence consists of its predicted answer and rationale. To obtain the student model’s target for a given instance, we append the gold answer to the teacher model’s input and use the teacher model’s resulting output. This guarantees that the teacher’s output includes the gold answer too. Using the teacher’s outputs as the student’s targets, we train two variants of the student model: (1) Rationalize-First, which first generates the rationale then predicts the answer; and (2) Predict-First, which first predicts the answer then generates the rationale. For a fair comparison, we used T5-Base as the backbone of the student model.
> >
> > The results below show the deficiencies of both Distilled Self-Rationalization variants.
> > - Rationalize-First’s predicted answer is strongly conditioned on the predicted rationale, so the predicted rationale serves as a crucial bottleneck for answer prediction. However, since the small student LM’s rationalization is merely supervised by the large teacher LM’s rationales, the student only has a noisy approximation of the teacher’s rationales at inference time. This hurts the student’s ability to predict the correct answer and causes Rationalize-First to generally underperforms other methods in accuracy. Our finding is corroborated by prior works, as the Rationalize-First paradigm has been shown to require very large LMs (over 100B parameters) to work effectively [1, 2].
> >
> > - Predict-First is trained to generate its predicted answer first, so its answer prediction does not rely on an LM to provide good rationales as context. Thus, it performs better than Rationalize-First, but still much worse than PINTO (which does have access to good rationales). However, since Predict-First’s rationales are generated after answer prediction, Predict-First can learn to generate the answer without considering the generated rationale. This severely undermines the rationales’ faithfulness, causing Predict-First to yield much lower LAS scores than other methods (esp. on StrategyQA).
> >
> > | Method | CSQA || StrategyQA || OBQA || QASC ||
> > | --- | ---: | ---: | ---: | ---: | ---: | ---: | ---: | ---: |
> > | | Acc. | LAS | Acc. | LAS | Acc. | LAS | Acc. | LAS |
> > | Distilled Rationalize-First | 51.97 | - | 50.20 | 1.29 | 48.90 | - | 33.34 | 31.82 |
> > | Distilled Predict-First | 55.77 | - | 54.61 | -2.68 | 50.25 | 12.30 | 34.53 | 18.48 |
> > | PINTO | **61.67** | **24.22** | **60.37** | **3.35** | **58.85** | **18.02** | **37.82** | **38.98** |
> >
> > [1] Wei et al. “Emergent abilities of large language models.” 2022.
> >
> > [2] Wei et al. “Chain of thought prompting elicits reasoning in large language models.” 2022.

---

> > > ### Author Response · Authors · 2022-11-19
> > > **Response to Reviewer gqDU (Part 3)**
> > >
> > > > 3. What is the quality of the generated rationales?
> > >
> > > Thank you for the suggestion! To investigate this, we conducted a user study on CSQA in which humans annotators evaluated the quality of both generated rationales and gold rationales, without knowing how the rationales were produced. Note that the gold rationales are already provided in the CSQA dataset. In our user study, we considered a random sample of 100 CSQA test instances, with each instance having two rationales (i.e., generated and gold). Also, our user study involved 3 annotators, all of whom are computer science undergraduate or graduate students.
> > >
> > > Following the setup in [1], annotators were asked to score each rationale along five dimensions, on a three-point Likert scale:
> > > 1. Factuality: Is the information provide by the explanation factual?
> > > 2. Grammaticality: Is the explanation grammatical?
> > > 3. Novelty: Does the explanation provide new facts, reasoning, or other information not stated in the corresponding question and answer?
> > > 4. Relevance: Is the explanation relevant for supporting the answer?
> > > 5.  Completeness: Does the explanation provide enough information to justify the answer?
> > >
> > > The table below presents our user study results, averaged over the 3 annotators. To measure inter-annotator agreement, we used Fleiss’ kappa and obtained K=0.34, indicating fair agreement. Our results show that the quality of generated rationales is very competitive with that of gold rationales. For factuality, grammaticality, novelty, and relevance, both rationale types yield about the same scores. In particular, generated rationales even slightly outperform gold rationales on grammaticality. However, for completeness, gold rationales noticeably outperform generated rationales. We believe this is due to the fact that CSQA’s human annotators were explicitly encouraged by the dataset creators to provide more comprehensive rationales [2].
> > >
> > >
> > > | Rationale Type | Factuality | Grammaticality | Novelty | Relevance | Completeness |
> > > | --- | ---: | ---: | ---: | ---: | ---: |
> > > | Generated | 0.91 | 0.99 | 0.69 | 0.87 | 0.65 |
> > > | Gold | 0.94 | 0.98 | 0.71 | 0.90 | 0.82 |
> > >
> > > [1] Wiegreffe et al. “Reframing human-ai collaboration for generating free-text explanations.” 2021.
> > >
> > > [2] Aggarwal et al. “Explanations for commonsenseqa: New dataset and models.” 2021.

---

> > > > ### Comment · Reviewer_gqDU · 2022-12-01
> > > > **Re: part 3**
> > > >
> > > > Thanks this is really helpful. I would compare it with davinci-002 generation, but I understand this might not be practical, due to a costly API.

---

> > > > > ### Author Response · Authors · 2022-12-03
> > > > > **Part 3: Discussion of User Study**
> > > > >
> > > > > > Thanks this is really helpful. I would compare it with davinci-002 generation, but I understand this might not be practical, due to a costly API.
> > > > >
> > > > > Thank you for your positive feedback and suggestion! We agree that expanding our user study to include davinci-002 rationales could be interesting, so we plan to add this in an updated version of our paper. Furthermore, we would like to emphasize that human-annotated gold rationales are generally assumed to be higher quality than any machine-generated rationales, so it is very encouraging to see that PINTO’s GPT-NeoX generated rationales already achieve similar quality to gold rationales.

---

> > > > > > ### Author Response · Authors · 2022-12-07
> > > > > > **Part 3: Addition of GPT-3 davinci-002 Results to User Study**
> > > > > >
> > > > > > Dear Reviewer gqDU,
> > > > > >
> > > > > > Following your helpful suggestion, we have expanded our user study to also include rationales generated by GPT-3 davinci-002. For GPT-3 rationales, we obtain a Fleiss’ kappa score of K=0.36, indicating fair agreement among the same three annotators.
> > > > > >
> > > > > > In our user study results below, GPT-NeoX slightly outperforms GPT-3 on Factuality, Grammaticality, and Relevance, whereas GPT-3 outperforms GPT-NeoX slightly on Novelty but considerably on Completeness. Thus, it seems that Completeness is the biggest distinguishing factor between the three rationale types, where achieving higher Completeness scores may require much higher annotation effort or model capacity.
> > > > > >
> > > > > > Overall, our user study demonstrates that PINTO’s GPT-NeoX rationales have similar quality as GPT-3 rationales and gold rationales, while requiring much lower annotation effort (vs. gold) or model capacity (vs. GPT-3). This validates our use of GPT-NeoX rationales as context for PINTO’s reasoning module.
> > > > > >
> > > > > >
> > > > > > | Rationale Type | Factuality | Grammaticality | Novelty | Relevance | Completeness |
> > > > > > | --- | ---: | ---: | ---: | ---: | ---: |
> > > > > > | GPT-NeoX | 0.91 | 0.99 | 0.69 | 0.87 | 0.65 |
> > > > > > | GPT-3 | 0.89 | 0.98 | 0.71 | 0.86 | 0.74 |
> > > > > > | Gold | 0.94 | 0.98 | 0.71 | 0.90 | 0.82 |
> > > > > >
> > > > > > We would love to hear your thoughts about whether our rebuttal sufficiently addresses your concerns and questions. If so, it would be great if you could update your score. Thank you so much!
> > > > > >
> > > > > > Sincerely,
> > > > > >
> > > > > > Paper5818 Authors

---

> > > ### Comment · Reviewer_gqDU · 2022-12-01
> > > **Re: Part 2**
> > >
> > > Thanks for running these experiments. If I read this correctly, the performance are not bad given the  38.41ACC of Prompted Self-Rationalization model (GPT-NeoX). No? Like based on Table 2, the gap between GPT-NeoX and Distilled is not big. Am I reading this correclty?

---

> > > > ### Author Response · Authors · 2022-12-03
> > > > **Part 2: Analysis of PINTO Baselines and Design Choices**
> > > >
> > > > > Thanks for running these experiments. If I read this correctly, the performance are not bad given the 38.41ACC of Prompted Self-Rationalization model (GPT-NeoX). No? Like based on Table 2, the gap between GPT-NeoX and Distilled is not big. Am I reading this correclty?
> > > >
> > > > Our results show that Distilled Self-Rationalization generally outperforms Prompted Self-Rationalization (GPT-NeoX), often by a large margin. For example, on CSQA, Distilled Rationalize-First and Distilled Predict-First achieve 51.97 and 55.77 accuracy, respectively, while Prompted Self-Rationalization (GPT-NeoX) only achieves 38.41 accuracy. We see similar trends on OpenBookQA and QASC, though Prompted Self-Rationalization (GPT-NeoX) actually performs better on StrategyQA.
> > > >
> > > > Recall that the Distilled Self-Rationalization models are finetuned to jointly generate an answer and rationale, supervised by the gold answers and GPT-NeoX’s rationales. The fact that Distilled Self-Rationalization outperforms Prompted Self-Rationalization (GPT-NeoX) justifies our design choice of finetuning with gold answer supervision.
> > > >
> > > > However, Standard Training and Dropout Context generally outperform both Distilled Self-Rationalization and Prompted Self-Rationalization (GPT-NeoX). This suggests the effectiveness of decoupling the model into dedicated rationalizing and reasoning modules, so that a single model does not have to be responsible for both rationalizing and reasoning.
> > > >
> > > > Ultimately, PINTO consistently outperforms all baselines. First, this further validates the effectiveness our aforementioned design choices: (1) finetuning with gold answer supervision and (2) decoupling the model into dedicated rationalizing and reasoning modules. Second, this justifies our use of counterfactual regularization to ensure that the reasoning module utilizes the generated rationales in a faithful/generalizable manner. Note that counterfactual regularization can only be used for finetuning-based methods, which poses another disadvantage of Prompted Self-Rationalization.

---

> > ### Comment · Reviewer_gqDU · 2022-12-01
> > **Re: part1**
> >
> > Thanks for listing your contribution. I still believe the overall technical contribution of this paper is marginal:
> >
> > > First, unlike fine-tuned self-rationalization and pipeline rationalization, PINTO does not require human-annotated rationale to train the rationalizing module.
> >
> > It is worth mentioning that the "Prompted Self-Rationalization" does not require the human rational either.
> >
> > > Second, unlike prompted self-rationalization, PINTO does not require the rationalizing module to be extremely large (i.e., over 100B parameters).
> >
> > It's worth mentioning "Prompted Self-Rationalization" does not require a further fine tuning step of <~1B model which can be expensive too, and it generate a task specific model that requires two steps.
> >
> > > Third, none of these prior works consider whether the generated rationales are faithful to the model’s reasoning process. Without any way to ensure rationale faithfulness, it is doubtful that prior works’ generated rationales are reliable for understanding the model’s decision-making, especially if the model is behaving problematically. To address this issue, PINTO’s counterfactual regularization encourages the reasoning module’s behavior to align with the generated rationale. We provide support for PINTO’s faithfulness in Sec. 5.2 and Fig. 6, where we show that the strong connection between PINTO’s rationalizing and reasoning enables it to achieve larger performance gains when its rationales are augmented with human feedback, compared to baselines without counterfactual regularization.
> >
> > Sure, this is a good contribution.
> >
> >
> > > Comparison to SOTA Methods
> >
> > Thanks this helps to have a better understanding of the generality of the model.

---

> > > ### Author Response · Authors · 2022-12-03
> > > **Part 1: PINTO’s Novelty/Contributions**
> > >
> > > > It is worth mentioning that the "Prompted Self-Rationalization" does not require the human rational either.
> > >
> > > We would like to clarify that **prior works only address at most one of the three limitations** we identify (i.e., rationale annotation effort, large LM requirement, rationale faithfulness). Meanwhile, **PINTO is the only method that jointly addresses all three limitations**, providing an effective tradeoff between rationale annotation effort and large LM requirement, with respect to both task performance and rationale faithfulness.
> > >
> > > > It's worth mentioning "Prompted Self-Rationalization" does not require a further fine tuning step of <~1B model which can be expensive too, and it generate a task specific model that requires two steps.
> > >
> > > We believe prompting a large (>100B) LM is much more computationally prohibitive than finetuning a small (~1B) LM. Very few users have the compute resources to even perform inference using a large LM with batch size 1, but it is quite common/accessible for users to finetune small LMs. Plus, whereas the typical upper bound on the number of forward/backward passes during training is relatively low (our finetuned LMs are only trained for 10 epochs), LMs are expected to perform arbitrarily many forward passes during inference. As a result, this scale disparity between small LMs and large LMs becomes extremely critical in real-world applications, such that large LMs are often infeasible to deploy.
> > >
> > > > Sure, this is a good contribution. … Thanks this helps to have a better understanding of the generality of the model.
> > >
> > > Thank you for your positive feedback! We are happy to hear that you appreciate our proposed counterfactual regularization strategy as well as our comparison to SOTA methods.

---

> ### Author Response · Authors · 2022-11-24
> **Friendly reminder to respond to author rebuttal**
>
> Dear Reviewer gqDU,
>
> Thank you again for your review! We are happy to hear that you appreciated PINTO’s experiments.
>
> Based on your thoughtful feedback, we wrote a detailed rebuttal covering the following points:
> - PINTO’s novelty and contributions
> - Comparison between PINTO and SOTA methods
> - Addition of Distilled Self-Rationalization baselines
> - Quality of PINTO’s generated rationales
>
> We would love to hear your thoughts about our rebuttal, including whether it sufficiently addresses your concerns and questions. If you believe that our rebuttal is satisfactory, it would be great if you could consider increasing your score. Any feedback is welcome and greatly appreciated!
>
> Sincerely,
>
> Paper5818 Authors

---

> ### Author Response · Authors · 2022-11-30
> **Friendly follow-up reminder to respond to author rebuttal**
>
> Dear Reviewer gqDU,
>
> Just wanted to follow up on our previous reminder message!
>
> We would love to hear your thoughts about our rebuttal, including whether it sufficiently addresses your concerns and questions. If you believe that our rebuttal is satisfactory, it would be great if you could consider increasing your score. Any feedback is welcome and greatly appreciated!
>
> Sincerely,
>
> Paper5818 Authors

---

> ### Author Response · Authors · 2022-12-03
> **Response to Reviewer gqDU's Rebuttal Comments**
>
> Dear Reviewer gqDU,
>
> We appreciate your thoughtful comments on our rebuttal! In response, we have provided additional clarification about the following items:
> - **Part 1: PINTO’s Novelty/Contributions** (https://openreview.net/forum?id=WBXbRs63oVu&noteId=OXc5C796SK)
> - **Part 2: Analysis of PINTO Baselines and Design Choices** (https://openreview.net/forum?id=WBXbRs63oVu&noteId=inMRidfKF5)
> - **Part 3: Discussion of User Study** (https://openreview.net/forum?id=WBXbRs63oVu&noteId=yL37s5Km0H_)
>
> If you feel that our latest response has addressed your concerns, it would be great if you could consider increasing your score. Thank you for your consideration!
>
> Sincerely,
>
> Paper5818 Authors

---

> > ### Author Response · Authors · 2022-12-06
> > **Friendly reminder to respond to author follow-up rebuttal**
> >
> > Dear Reviewer gqDU,
> >
> > Just wanted to follow up on our previous message!
> >
> > We would love to hear your thoughts about our follow-up rebuttal (https://openreview.net/forum?id=WBXbRs63oVu&noteId=1XauL9v-eHI), including whether it sufficiently addresses your concerns and questions. If you believe that our follow-up rebuttal is satisfactory, it would be great if you could consider increasing your score. Any feedback is welcome and greatly appreciated!
> >
> > Sincerely,
> >
> > Paper5818 Authors

---

> > ### Author Response · Authors · 2022-12-09
> > **Another friendly reminder to respond to author follow-up rebuttal**
> >
> > Dear Reviewer gqDU,
> >
> > Just wanted to follow up on our previous message! As you know, we are quickly approaching the Discussion Stage 2 deadline on December 12.
> >
> > In light of this, we would love to hear your thoughts about our follow-up rebuttal (https://openreview.net/forum?id=WBXbRs63oVu&noteId=1XauL9v-eHI), including whether it sufficiently addresses your concerns and questions.
> >
> > In particular, we followed your suggestion to include GPT-3 davinci-002 rationales in our user study on rationale quality (https://openreview.net/forum?id=WBXbRs63oVu&noteId=gRK20rtfy4), showing that PINTO's GPT-NeoX rationales are similar in quality to GPT-3 and gold rationales.
> >
> > If you believe that our follow-up rebuttal is satisfactory, it would be great if you could consider increasing your score. Any feedback is welcome and greatly appreciated!
> >
> > Sincerely,
> >
> > Paper5818 Authors

---

### Official Review · Reviewer_mr1Y · 2022-11-03

**Confidence:** 4
**Correctness:** 3
**Technical Novelty And Significance:** 3
**Empirical Novelty And Significance:** 2
**Recommendation:** 8

**Clarity, Quality, Novelty And Reproducibility:**

- The paper is well written, appropriately organized, and easy to follow
- The method is novel as far as I am aware
- The method is straightforward and practical
- The general quality is good, but some limitations remain (will increase the scores if the authors can address them)

**Strength And Weaknesses:**

Strengths:

- The method is simple, sound, and makes intuitive sense. To my knowledge, this is the first work to
- The general evaluation setup is largely appropriate such as choice of datasets, ID vs OOD evaluation and measuring faithulness

Weaknesses:

- The current formulation only allows the method to be applied to multi-choice classification problem or where the classification target space is not too large
- The baselines are not on the same parameter scale especially the w/o Rationalization baseline. It is unclear whether the performance gain is coming from using a strong rationale LM or that this framework in-general is better than the other options. (This might explain why ID even increases.)
- One way control for the issue above might be using rationale modules of different sizes, and use a small LM (<1B) to fine-tune on the rationales generated by these different modules, and eventually stack it with the proposed PINTO training procedure. At the very least it might be a better comparison to the Fine-tuned self-rationalization baseline.
- It might be a stretch to draw any conclusion from Figure 4 as mentioned in section 5.2 due to the high variance and lack of trend
- The issue of answer leakage is mentioned in section 3, I am not sure how the proposed framework circumvent such an issue. With PINTO, the rationale can still contain answer information right? (Please correct me if I misunderstood.)
- Would be nice to have variance statistics

**Summary Of The Paper:**

The paper proposes an LM pipeline that first generate free-text rationales and use the generated rationales to fine-tune a reasoning module such that it makes the prediction that relies on the rationale as much as possible. A regularization scheme is proposed to mitigate issue where rationales might be ignored.

The method is evaluated on in-domain and out-of-domain setups using accuracy and faithfulness as metrics.

**Summary Of The Review:**

The paper presents a practical method to solve multi-choice classification problem using rationales. The paper is well-written and provides a new way of incorporating rationales to improve OOD performance. Despite obvious limitations and some issues in how the baselines are constructed, there is no major flaws.

---

> ### Author Response · Authors · 2022-11-19
> **Response to Reviewer mr1Y**
>
> Thank you for your positive review and very helpful comments! Below are our responses and we have updated the draft accordingly.
>
> > 1. The current formulation only allows the method to be applied to multi-choice classification problem.
>
> We acknowledge that PINTO’s choice-specific (i.e., class-specific) counterfactual regularization requires the task to have a finite label space per instance. In our paper, we consider multi-choice QA, in which each instance has a different but fixed-size set of classes. We specifically focus on multi-choice QA because it is the most prevalent problem setting in the commonsense/implicit reasoning literature, such that existing baseline methods are most commonly evaluated on multi-choice QA [1, 2]. However, PINTO could also be applied to other classification tasks where all instances share the same fixed-size set of classes (e.g., sentiment analysis, NLI). In future work, we are interested in investigating how PINTO-like methods can be applied to generation tasks and other classification tasks.
>
> [1] Lourie et al. "Unicorn on rainbow: A universal commonsense reasoning model on a new multitask benchmark." 2021.
>
> [2] Storks et al. "Commonsense reasoning for natural language understanding: A survey of benchmarks, resources, and approaches." 2019.
>
> > 2. Use rationale modules of different sizes, and use a small LM (<1B) to fine-tune on the rationales generated by these different modules.
>
> Thank you for the suggestion! In addition to using GPT-NeoX (20B) as the rationalizing module, we have now added CSQA results by using GPT-2 (1.5B), GPT-J (6B), and GPT-3 (175B) as the rationalizing module. Since GPT-3 has high compute costs and is not freely available, we only used it to generate rationales for test instances at inference time, with GPT-NeoX’s generated rationales used during training. Note that all results here are obtained using T5-Base (220M) as the reasoning module. Across all four rationalizing module architectures, we find that PINTO outperforms the Standard and Dropout baselines. This suggests that PINTO’s performance gains are agnostic to the size of the rationalizing module.
>
> | LM type | GPT-2 (1.5B) | GPT-J (6B) | GPT-NeoX (20B) | GPT-3 (175B) |
> | --- | --- | --- | --- | --- |
> | Standard | 59.19 | 59.44 | 59.48 | 59.40 |
> | Dropout | 58.82 | 58.48 | 59.64 | 58.64 |
> | PINTO | **61.10** | **60.73** | **61.67** | **61.32** |
>
> > 3. With PINTO, the rationale can still contain answer information right?
>
> Yes. As we discussed in Sec. 3, PINTO’s rationales are specifically designed to be choice-specific. As a result, the reasoning module is encouraged to predict the correct answer choice based on which choice’s rationale makes the most sense. We opted for this rather than generating a single choice-agnostic rationale, because in that case the rationalization module is forced to make an “early decision” about the predicted answer.
>
> > 4. Would be nice to have variance statistics
>
> Due to space constraints, we have added the variance statistics to Table 9-10 in the appendix.

---

> ### Author Response · Authors · 2022-11-24
> **Friendly reminder to respond to author rebuttal**
>
> Dear Reviewer mr1Y,
>
> Thank you again for your review! We are happy to hear that you appreciated PINTO’s novelty, performance/efficiency improvements, experiments, motivation, and presentation.
>
> Based on your thoughtful feedback, we wrote a detailed rebuttal covering the following points:
> - PINTO’s general applicability to classification tasks
> - PINTO’s effectiveness across various scales of the rationalizing module
> - Intuition behind PINTO’s rationalizing/reasoning design
> - Addition of variance statistics for ID and OOD results
>
> We would love to hear your thoughts about our rebuttal, including whether it sufficiently addresses your concerns and questions. If you believe that our rebuttal is satisfactory, it would be great if you could consider increasing your score. Any feedback is welcome and greatly appreciated!
>
> Sincerely,
>
> Paper5818 Authors

---

> ### Author Response · Authors · 2022-11-30
> **Friendly follow-up reminder to respond to author rebuttal**
>
> Dear Reviewer mr1Y,
>
> Just wanted to follow up on our previous reminder message!
>
> We would love to hear your thoughts about our rebuttal, including whether it sufficiently addresses your concerns and questions. If you believe that our rebuttal is satisfactory, it would be great if you could consider increasing your score. Any feedback is welcome and greatly appreciated!
>
> Sincerely,
>
> Paper5818 Authors

---

> > ### Comment · Reviewer_mr1Y · 2022-11-30
> > **Thank you for addressing the concerns. I'm happy to raise the score to 8.**
> >
> > Thanks for the detailed explanations to my questions. In particular, the added CSQA results on different model sizes addresses my main concern. Other explanations also cleared my confusions. I think the work is solid and the evaluation/analysis is appropriate. I'm comfortable to raise the score to 8.

---

> > > ### Author Response · Authors · 2022-11-30
> > > **Thank you for your feedback! Friendly reminder about updating official score.**
> > >
> > > Dear Reviewer mr1Y,
> > >
> > > Thanks for your positive feedback! We are happy to hear that our rebuttal addressed your concerns and that you've decided to increase your score to 7.
> > >
> > > In light of this, we have one small favor to ask. Currently, your official score is still showing as 6 in your original review (https://openreview.net/forum?id=WBXbRs63oVu&noteId=FKYV7Q8aaI). To make your new score more visible to the ICLR chairs, it would be great if you could also update your official score to 7.
> > >
> > > Thank you so much for your time!
> > >
> > > Sincerely,
> > >
> > > Paper5818 Authors

---

> > > > ### Comment · Reviewer_mr1Y · 2022-11-30
> > > > **Score updated.**
> > > >
> > > > I have updated the score to 8. Thank you for the reminder.

---

> > > > > ### Author Response · Authors · 2022-12-01
> > > > > **Thank you!**
> > > > >
> > > > > Thank you for updating your score!

---

> > > > > > ### Comment · Reviewer_mr1Y · 2022-12-09
> > > > > > **Post-reviewer discussion scoring.**
> > > > > >
> > > > > > Dear authors,
> > > > > >
> > > > > > After the discussion between the reviewers, I decide to keep the original score (post-rebuttal), which is 8.

---

### Official Review · Reviewer_8SKc · 2022-11-07

**Confidence:** 4
**Correctness:** 3
**Technical Novelty And Significance:** 3
**Empirical Novelty And Significance:** 2
**Recommendation:** 6

**Clarity, Quality, Novelty And Reproducibility:**

- Clarity: The paper is well-written, and I can understand technical details.
- Quality: The paper is solid with clear motivation and detailed empirical results.
- Novelty: The paper is novel for its regularization on reasoning module.
- Reproducibility: Some hyperparameters are missing but one should be able to reproduce with some effort.

**Strength And Weaknesses:**

Strength:
- The paper proposes a novel LM pipeline that rationalizes with prompt-based learning and reasons via counterfactual regularization.
- The empirical results show the advantage of the design choice: 1) pipeline-based model over unified model and 2) regularization method for reasoning module
- The paper is well-written

Weakness:
- The evaluation is only about the proposed method and its own variants (baselines). But the performance is much lower than the state-of-the-art results. For instance, the best performing CSQA system has accuracy around 80% (https://www.tau-nlp.sites.tau.ac.il/csqa-leaderboard2), and the proposed method only achieves 60%. Of course, the best systems use additional knowledge sources with careful task-specific tuning. But the gap makes me wonder if the proposed method could help upon the state-of-the-art system.


**Summary Of The Paper:**

The paper proposes a pipelined method for reasoning tasks with language models. The proposed pipeline has two parts: a rational generation module and a reasoning module. The proposed method has two major novelties: 1) the rationale generation uses a frozen LM with few-shot prompts. 2) The reasoning module uses regularizations to improve the faithfulness of utilizing the rationales.

The paper compares the proposed method with several baseline methods and shows that the proposed model achieves better performance on 4 QA tasks which require external knowledge. They also test the out-of-distribution performance and robustness of noisy rationales of their proposed method.

**Summary Of The Review:**

The paper proposes an interesting method combining LMs with different capacities for open-domain QA tasks. The first LM generates answer-specific rationales for the second LM to reason about. Two regularization objectives are proposed to improve faithfulness. The ablation study on 4 widely used benchmark shows promising performance of the proposed method. However, the performance is far away from the state-of-the-art system for the corresponding tasks.

---

> ### Author Response · Authors · 2022-11-19
> **Response to Reviewer 8SKc**
>
> Thank you for your helpful comments and insightful questions! Based on your feedback, we have updated the draft and provided the responses below.
>
> > 1. The performance is much lower than the state-of-the-art results. The gap makes me wonder if the proposed method could help upon the state-of-the-art system.
>
> In our submission, we focused on T5-Base as the reasoning module. However, PINTO can also accommodate larger reasoning module architectures, which are critical to achieving high task performance. Below, we show CSQA and OBQA results by using SOTA LM architectures (i.e., RoBERTa-Large and T5-3B) as the reasoning module. As shown in the results below, using either RoBERTa-Large or T5-3B, PINTO still outperforms the baselines in both task performance (Acc.) and faithfulness (LAS). In particular, PINTO with T5-3B yields SOTA performance on both CSQA and OBQA. This suggests that PINTO is generally effective across different LM architectures.
>
> | Method with RoBERTa-Large | CSQA || OBQA ||
> | --- | ---: | ---: | ---: | ---: |
> | | Acc. | LAS | Acc. | LAS |
> | w/o Rationale | 73.57 | - | 66.4 | - |
> | Standard | 73.98 | 29.50 | 69.9 | 26.04 |
> | Dropout | 71.81 | **​​30.13** | 68.6 | 25.06 |
> | PINTO | **74.73** | 29.36 | **72.6**| **26.72** |
>
> | Method with T5-3B | CSQA || OBQA ||
> | --- | ---: | ---: | ---: | ---: |
> | | Acc. | LAS | Acc. | LAS |
> | w/o Rationale | 80.94 | - | 73.6 | - |
> | Standard | 81.70 | 28.22 | 79.1 | 22.34 |
> | Dropout | 81.35 | 26.90 | 78.2 | 22.13 |
> | PINTO | **82.39** | **29.78** | **80.35** | **23.16** |

---

> ### Author Response · Authors · 2022-11-24
> **Friendly reminder to respond to author rebuttal**
>
> Dear Reviewer 8SKc,
>
> Thank you again for your review! We are happy to hear that you appreciated PINTO’s novelty, performance/efficiency improvements, experiments, motivation, and presentation.
>
> Based on your thoughtful feedback, we wrote a detailed rebuttal covering the following points:
> - Comparison between PINTO and SOTA methods
>
> We would love to hear your thoughts about our rebuttal, including whether it sufficiently addresses your concerns and questions. If you believe that our rebuttal is satisfactory, it would be great if you could consider increasing your score. Any feedback is welcome and greatly appreciated!
>
> Sincerely,
>
> Paper5818 Authors

---

> ### Author Response · Authors · 2022-11-30
> **Friendly follow-up reminder to respond to author rebuttal**
>
> Dear Reviewer 8SKc,
>
> Just wanted to follow up on our previous reminder message!
>
> We would love to hear your thoughts about our rebuttal, including whether it sufficiently addresses your concerns and questions. If you believe that our rebuttal is satisfactory, it would be great if you could consider increasing your score. Any feedback is welcome and greatly appreciated!
>
> Sincerely,
>
> Paper5818 Authors

---

> > ### Comment · Reviewer_8SKc · 2022-12-02
> > **Updated score for the new results**
> >
> > Thanks for the updated result. The gap between the proposed method and the SOTA ones is smaller now. I updated my score accordingly.

---

> > > ### Author Response · Authors · 2022-12-02
> > > **Thank you for your feedback!**
> > >
> > > Dear Reviewer 8SKc,
> > >
> > > Thank you for your positive feedback! We are happy to hear that our rebuttal addressed your concerns and that you've decided to increase your score to 6.
> > >
> > > Sincerely,
> > >
> > > Paper5818 Authors

---

### Author Response · Authors · 2022-11-19
**General Response**

We thank all of the reviewers for their thoughtful feedback and recognition of our paper’s contributions! We are delighted to see that the reviewers appreciated PINTO’s novelty (8SKc, mr1Y, WxQk), performance/efficiency improvements (8SKc, mr1Y, WxQk), experiments (8SKc, mr1Y, gqDU, WxQk), motivation (8SKc, mr1Y, WxQk), and presentation (8SKc, mr1Y, WxQk).

In response to the reviewers’ comments/questions, we have addressed the following items in our rebuttal and updated paper:
- Demonstrated that PINTO can achieve **SOTA task performance**, by expanding our experiments to also consider stronger LM architectures (RoBERTa-Large, T5-3B) for the reasoning module. These results are in Tables 6-7 of the appendix. **[8SKc, gqDU, WxQk]**
- Showed that PINTO consistently outperforms all baselines across **various scales of the rationalizing module**, by presenting an ablation study comparing the use of GPT-2 (1.5B), GPT-J (6B), and GPT-3 (175B) as the rationalizing module. These results are in Table 8 of the appendix. **[mr1Y]**
- Added as baselines two variants of Distilled Self-Rationalization, which supervises a small Self-Rationalizing model (T5-Base) using the outputs generated by a large Prompted Self-Rationalization model (GPT-NeoX). PINTO consistently outperforms these baselines in both task performance and faithfulness. These results are highlighted in red in Tables 2-3. **[gqDU]**
- Conducted a user study to show that PINTO’s generated rationales have similar quality as (human-annotated) gold rationales. This experiment is in Sec. A.4 of the appendix. **[gqDU]**
- Added variance statistics for the reported results to Table 9-10 of the appendix. Note that we bolded the results that exceed the second-best result with statistical significance in Table 2-3 already. **[mr1Y]**
- Summarized PINTO’s novelty and distinction from existing paradigms (below). We will update the paper to better convey this. **[gqDU]**
- Discussed PINTO’s general applicability to classification tasks. **[mr1Y]**
- Discussed the intuition behind the PINTO’s rationalizing/reasoning design. **[mr1Y]**
- Discussed the connection between PINTO’s rationale faithfulness (via counterfactual regularization) and its OOD generalization ability. **[WxQk]**
- Discussed how PINTO’s faithful rationalization impacts its robustness to perturbed rationales. **[WxQk]**
- Discussed the effects of prompt design on both rationale quality and PINTO’s performance. **[WxQk]**
- Made several writing updates, including: (1) adding the description and the results of the two Distilled Self-Rationalization baselines in Sec. 4, (2) moving the results of Prompted Self-Rationalization (GPT-3) to Table 2, and (3) reinterpreting and reorganizing the results shown in Table 4.

**Summary of Contributions**
1. We propose **PINTO**, a novel LM pipeline that rationalizes via prompt-based learning and learns to faithfully reason over the generated rationales via counterfactual regularization.
- **PINTO’s rationalizing module** prompts a medium-scale LM to generate free-text rationales. Unlike prior works, PINTO does not require expensive gold rationale annotations for training the rationalizing module, since the rationalizing module is prompted instead of fine-tuned. Also, PINTO does not require the rationalizing module to be a large-scale LM with 100B+ parameters, since a separate reasoning module is used to solve the task.
- **PINTO’s reasoning module** is trained to solve the task, using the rationalizing module’s generated rationale as context. To ensure that its behavior is properly dictated by the generated rationale, we propose training the reasoning module with counterfactual regularization. This encourages the model to produce less confident predictions when the rationale is noisily perturbed. As a result, the generated rationale can serve as a more faithful explanation of the reasoning module’s behavior.
2. PINTO achieves **SOTA performance** on both **task performance** and rationale **faithfulness** evaluations, yielding significant improvements over various baselines.
- We demonstrate this through comprehensive experiments on both ID and OOD test sets from four popular question answering datasets (CSQA, StrategyQA, OBQA, QASC).
- We show that PINTO consistently outperforms baselines across a wide range of rationalizing (T5-Base, GPT-2, GPT-J, GPT-3) and reasoning (T5-Base, RoBERTa-Large, T5-3B) module architectures.
- We conduct a user study to show that PINTO’s generated rationales have similar quality as gold rationales, which helps validate the reliability of PINTO’s rationalizing module.

---

### Decision · Program_Chairs · 2023-01-20

**Decision:**

Accept: poster

**Justification For Why Not Higher Score:**

See reviewer discussion above

**Justification For Why Not Lower Score:**

The benefits of the paper seem to slightly outweigh the weaknesses.

**Metareview: Summary, Strengths And Weaknesses:**

This paper proposes a method (PINTO) for question answering via generating free-form text rationales using language models. The key difference with prior work that has also explored similar setups is the fact that this paper uses LM prompting to generate rationales for each answer choice (without any finetuning) and then fine-tunes a separate reasoning model that produces plausibility scores for all the answers along with their generated rationales. Experiments are performed on four QA datasets and although the proposed method does not outperform all baselines (e.g. GPT-3 w/ self-prompting) and is worse than SOTA (e.g. self-consistency for chain of thought, Wang et al., 2022), it uses less parameters overall and the authors also conduct several analyses that provide interesting findings. The authors also responded to several reviewer concerns around novelty and added new baselines to the paper based on reviewer suggestions. The reviewer discussion also raised some points around the clarity of the messaging and the scope of the proposed method.
If accepted, I encourage the authors to 1) more clearly specify that the key contribution/focus is on model efficiency (in terms of number of parameters), since other methods like self-prompting also do not require human-labeled rationales, and 2) more explicitly state the scope (in abstract and intro) to be choice-based question answering tasks.

**Note From Pc:**

if the above contains the word "oral" or "spotlight" please see: "oral" presentation means -> notable-top-5% and "spotlight" means -> notable-top-25%. As stated in our emails, we are disassociating presentation type from AC recommendations

**Summary Of Ac-Reviewer Meeting:**

Reviewer WxQk and 8SKc could not join.

The discussion centered around some key weaknesses brought up by reviewers - 1) incremental novelty compared to prior work, 2) the motivation of the paper was not clear, 3) the empirical results are not very impressive compared to the baselines and SOTA numbers. Specifically for 1), prior work has explored rationale generation through prompting (without human labels) and the key difference here seems to be that they use a smaller reasoning model while having a large model for prompted rational generation.

However, the positives were that the method uses a smaller parameter LM and runs several good human evaluations and analyses, which can be useful in informing future work in this direction.

Overall, while being borderline, the benefits of the paper's findings seem to slightly outweigh the drawbacks.